# Why Chain of Thought Fails in Clinical Text Understanding

## Abstract

Large language models (LLMs) are increasingly being applied to clinical care, a domain where both accuracy and transparent reasoning are critical for safe and trustworthy deployment. Chain-of-thought (CoT) prompting, which elicits step-by-step reasoning, has demonstrated improvements in performance and interpretability across a wide range of tasks. However, its effectiveness in clinical contexts remains largely unexplored, particularly in the context of electronic health records (EHRs), the primary source of clinical documentation, which are often lengthy, fragmented, and noisy. In this work, we present the first large-scale systematic study of CoT for clinical text understanding. We assess 95 advanced LLMs on 87 real-world clinical text tasks, covering 9 languages and 8 task types. Contrary to prior findings in other domains, we observe that 86.3% of models suffer consistent performance degradation in the CoT setting. More capable models remain relatively robust, while weaker ones suffer substantial declines. To better characterize these effects, we perform fine-grained analyses of reasoning length, medical concept alignment, and error profiles, leveraging both LLM-as-a-judge evaluation and clinical expert evaluation. Our results uncover systematic patterns in when and why CoT fails in clinical contexts, which highlight a critical paradox: *CoT enhances interpretability but may undermine reliability in clinical text tasks.* This work provides an empirical basis for clinical reasoning strategies of LLMs, highlighting the need for transparent and trustworthy approaches.

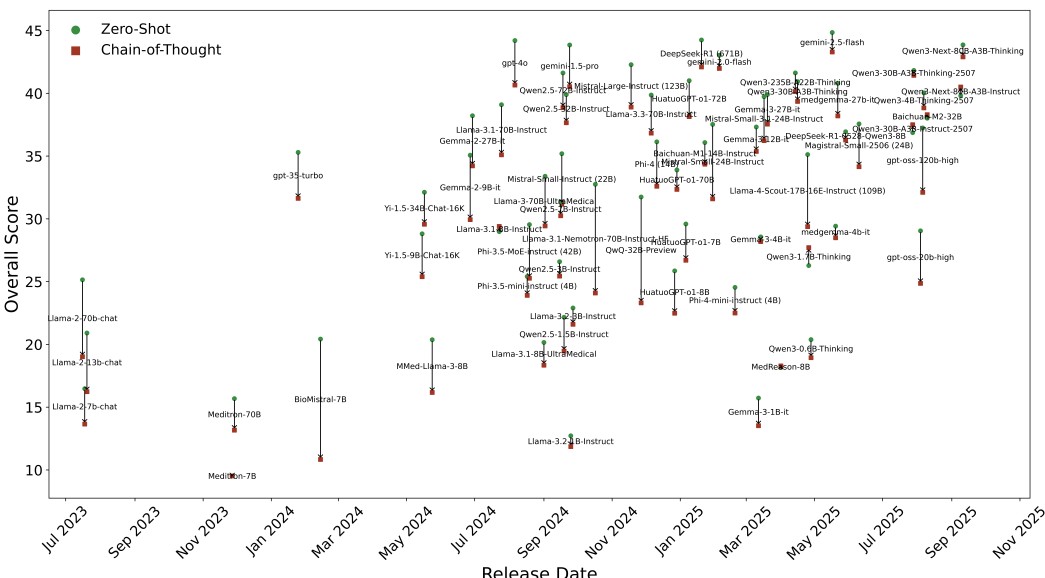

Figure 1: Performance of LLMs in clinical text understanding under Zero-shot and CoT.

## 1 INTRODUCTION

Large language models (LLMs) are increasingly being explored for clinical workflows, including clinical decision support, documentation drafting, information extraction, and telemedicine communication (Park et al., 2024; Vrdoljak et al., 2025). However, deploying LLMs in healthcare, an inherently safety-critical setting, requires both high accuracy and clear interpretability (Habli et al., 2020; Ennab & Mcheick, 2024). Providing explicit and transparent reasoning allows clinicians to critically evaluate LLM decision-making processes, identify gaps or misapplications of evidence, and make more informed, trustworthy decisions in patient care (Amann et al., 2020). Therefore, reliability and transparency are crucial for developing trustworthy clinical AI tools and their integration into clinical practice (Kim et al., 2025).

Chain-of-thought (CoT) prompting (Wei et al., 2022), which encourages language models to generate intermediate reasoning steps prior to producing a final answer, has become a widely adopted technique for exposing intermediate rationales. Using explicit instruction, e.g., *"Let's think step by step"*, CoT prompting can unlock the latent reasoning capabilities of LLMs, substantially improving performance and perceived interpretability across various domains. This effect is especially pronounced for tasks requiring advanced reasoning and logical coherence, such as mathematical problem-solving and coding (Kojima et al., 2022).

However, the effectiveness of CoT in clinical contexts remains underexplored and poorly understood. Existing evidence largely stems from exam-style or open-domain settings (e.g., USMLE-style QA) (Wu et al., 2024b; Liévin et al., 2024) rather than real-world, multi-source clinical narratives that demand heavy factual grounding and strict differential reasoning. The clinical text in electronic health records (EHRs) is noisy, ungrammatical, and highly variable, creating additional challenges for understanding. Moreover, previous studies have revealed that LLM outputs can contain hallucinations and incorrect information, potentially introducing patient-safety risks (Asgari et al., 2025; Roustan et al., 2025). Recent analyses further caution that CoT explanations may be unfaithful, representing superficially plausible narratives that do not reflect the model's actual decision basis, inviting over-trust if used as a reliability signal (Lyu et al., 2023). These open questions motivate a dedicated, large-scale assessment of CoT clinical text understanding.

In this study, we present the first large-scale investigation on the effectiveness of CoT prompting in LLMs in the understanding of clinical text. We evaluate 95 contemporary LLMs, including proprietary, open-source, and medical-domain variants, across 87 real-world clinical tasks spanning nine languages and eight task types, using standardized zero-shot and CoT inference protocols. Contrary to general-domain findings, we reveal that CoT consistently reduces performance despite increasing the apparent transparency of model outputs. To further investigate when and why CoT fails, we conduct four complementary analyses: (i) relating accuracy to reasoning trace length; (ii) quantifying clinical concept alignment between inputs and CoT traces; (iii) comparing lexical characteristics of CoT traces for correct vs. incorrect predictions; and (iv) auditing failure modes via an LLM-as-a-Judge with expert validation.

Overall, the main contributions of this work are:

- A comprehensive head-to-head comparison of CoT vs. zero-shot across 95 LLMs and 87 multilingual clinical tasks, covering various clinical NLP applications under a unified evaluation protocol.

- Empirical evidence demonstrating that CoT systematically undermines accuracy in clinical text understanding, with the larger drops for models of lower capability.

- Mechanistic analyses linking CoT failures to longer reasoning chains and weaker clinical-concept grounding, along with characterization of lexical signatures associated with incorrect answers (e.g., clinical abbreviations and numeric mentions).

- Development of an error taxonomy (hallucination, omission, incompleteness) evaluated via an LLM-as-a-Judge framework with clinician review, offering actionable guidance for safer use of CoT in the clinical context.

## 2 Related Work

### 2.1 Large Language Models and Chain-of-Thought

The large-scale pre-training and extensive fine-tuning equip LLM with rich world knowledge and logical reasoning capabilities, enabling them to generalize to broad tasks (Raiaan et al., 2024). Wei et al. (2022) demonstrated that CoT prompting can guide models to generate intermediate reasoning steps and significantly improve performance on tasks spanning arithmetic, commonsense, and symbolic reasoning. Then, a simple "*Let's think step by step*" instruction can elicit zero-shot reasoning Kojima et al. (2022), and self-consistency Wang et al. (2023) further boosts performance by aggregating diverse reasoning paths. These techniques have become a common practice for providing detailed explanations and improving performance. Subsequently, more advanced frameworks are built on CoT: ReAct interleaves reasoning with tool use (Yao et al., 2023b); Tree-of-Thoughts (Yao et al., 2023a) and Graph-of-Thoughts Besta et al. (2024) integrate structured planning and search with CoT; Least-to-Most (Zhou et al., 2023) decomposes complex problems into solvable subproblems. Furthermore, reinforcement-learning-based reasoning models (e.g., OpenAI o1 (Jaech et al., 2024) and DeepSeek-R1 (Guo et al.)) report additional gains by training models to "think before answering," though these advances are demonstrated mainly on math, coding, or synthetic reasoning benchmarks rather than safety-critical clinical text.

### 2.2 Medical Reasoning

The clinical applications of LLMs have rapidly emerged. Notably, Med-PaLM-2 (Singhal et al., 2025), integrating base-model improvements with prompting/ensemble strategies inspired by CoT, reports expert-level performance across several medical examinations. Meanwhile, In-Context Padding (Wu et al., 2024c) infers important medical entities from the input and uses them to steer the model's reasoning trajectory, improving alignment with clinical decision pathways. CLINICR (Nachane et al., 2024) mirrors the prospective process of incremental reasoning to support differential diagnosis. These methods aim to make intermediate steps clinically grounded rather than purely free-form rationales. However, Wu et al. (2024b) and Liévin et al. (2024) also revealed that CoT does not consistently improve the performance of general LLMs on medical questions. Especially, Zhao et al. (2025) demonstrated that the benefit of CoT reasoning vanishes when it is pushed beyond training distributions. However, there is still no systematic, large-scale evaluation of CoT on clinical text tasks. Furthermore, safety-oriented assessments in clinical documentation and summarization report hallucinations (Shah, 2024), such as fabricated facts, misattributed conditions, and instruction-following failures with potential patient-safety implications, underscoring the need for domain-specific evaluation protocols that go beyond exam-style QA and measure factual grounding in real-world clinical text (Asgari et al., 2025).

## 3 Methodology

### 3.1 Task Formulation

Let $\mathcal{T}$ denote a collection of clinical text tasks. For each task $t \in \mathcal{T}$, we consider a dataset $\mathcal{D}_t = \{(x_i, y_i)\}_{i=1}^{N_t}$, where $x_i$ is the clinical text that will serve as the model input, such as a clinical note or patient query, and $y_i$ is the task-specific reference output. Each task is paired with a detailed task instruction $p_t$ that specifies the clinical application scenario, the task requirement, and the expected output.

Given the concatenation of task instruction and input, an LLM $M$ is instructed to produce an output $\hat{y}_i$ according to the prompting protocol.

Specially, we investigate the model's capabilities under two prompting strategies:

- **Zero-shot:** The zero-shot prompt $p_t^{\mathrm{zs}}$ is a concise, task-specific instruction that asks the model to return only the final answer in the required format, without any explanation or auxiliary text. Formally, in the *zero-shot* setting, the model is

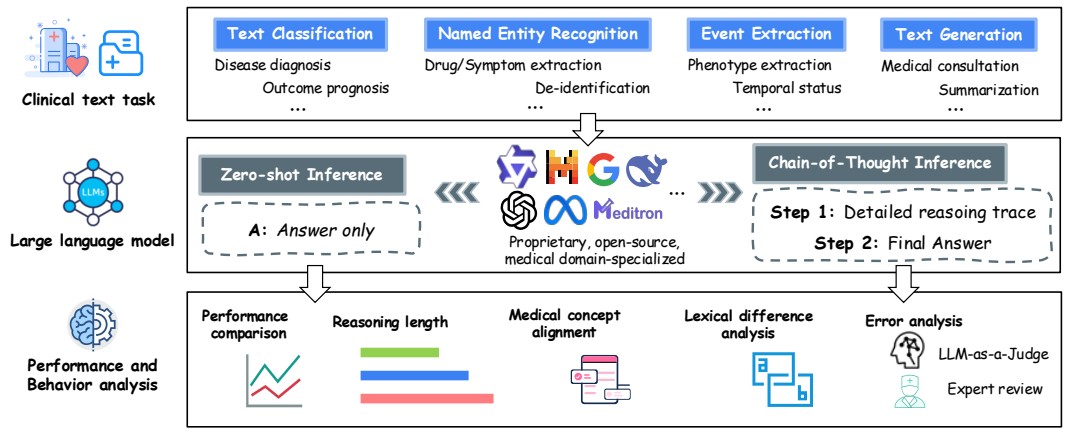

Figure 2: Evaluating CoT in clinical text understanding and probing why it fails.

instructed to output only the final answer for a given input:

$$\hat{y}_i^{\text{zs}} = M\big([\,p_t^{\text{zs}} \,\|\, x_i\,]\big).$$

No intermediate reasoning is requested or scored in this mode.

- **Chain-of-thought prompting:** Similar to zero-shot, the CoT prompt $p_t^{\text{cot}}$ includes the same task instruction and input, but adds an explicit request for step-by-step reasoning before the final answer. Formally, in the *chain-of-thought* (CoT) setting, the model is instructed to produce a stepwise reasoning trace before providing the final answer.

$$(r_i, a_i) = M\big([\,p_t^{\text{cot}} \,\|\, x_i\,]\big), \qquad \hat{y}_i^{\text{cot}} = a_i.$$

where $r_i$ is the free-form reasoning analysis generated by the model, and $a_i$ is the final answer; only $a_i$ is used for answer extraction and scoring.

## 3.2 Clinical Text Datasets

We include 87 clinical text datasets from the BRIDGE benchmark (Wu et al., 2024a; 2025). All tasks originate from real-world clinician–patient interactions drawn from electronic health records, clinical case reports, and online consultation records, with appropriate de-identification and license compliance.

The collection spans nine languages: English, Chinese, Spanish, Japanese, German, Russian, French, Norwegian, and Portuguese. They cover 8 clinical NLP tasks to support a broad scope of clinical applications, including text classification (*i.e.,* disease diagnosis), Named Entity Recognition (NER) and event extraction (*i.e.,* information extraction), and Question-Answering (QA). This breadth of languages, task types, and authentic clinical settings captures the diverse reasoning demands of practice and provides a rigorous substrate for analyzing the capabilities and limitations of CoT reasoning in clinical contexts.

## 3.3 Models and Evaluation Protocol

We investigate 95 widely used LLMs covering proprietary systems (*e.g.,* Google-Gemini, Openai-GPT-4o), open-source families (*e.g.,* Deepseek, Qwen, LLaMA, Mistral), and domain-specialized medical models (*e.g.,* Me-LLaMA, Meditron), with parameter scales from ∼1B to 671B. The complete model list is provided in the Appendix Table S1.

We compare performance under the two prompting strategies introduced in Sec.3.1. For each task, we leverage the task-specific scripts for label extraction and scoring provided by datasets, ensuring consistency and comparability across tasks and models.

- For text classification, semantic similarity, natural language inference, and document-level normalization and coding, the primary metric is accuracy.

- For NER, event extraction, and entity-level normalization and coding, we use event-level $F_1$ score, which requires the model to recover the target entity or event and all its associated attributes, measuring precision and recall jointly over structures.

- For QA and summarization, we report ROUGE-average, the micro average of ROUGE-1, ROUGE-2, and ROUGE-L, to capture overlap at token and sequence levels between model response and reference.

Formally, each task $t$ specifies a deterministic scoring function $S_t(\hat{y}, y) \in [0, 1]$. Given predictions under zero-shot and CoT for the same instances, we quantify the CoT effect for a task–model pair by:

$$\Delta Score = S_t^{\text{cot}} - S_t^{\text{zs}}, \qquad \Delta Score(\%) = \frac{S_t^{\text{cot}} - S_t^{\text{zs}}}{S_t^{\text{zs}}},$$

For each model $M$, the overall score is the macro-average of task-level scores across $\mathcal{T}$, yielding $\overline{S}_M^{\text{zs}}$ and $\overline{S}_M^{\text{cot}}$, and the corresponding aggregate effects $\overline{\Delta Score}_M$ and $\overline{\Delta Score(\%)}_M$.

### 3.4 Implementation Details

**Prompt design.** To examine LLMs under broadly adopted settings, the CoT prompt follows the classical CoT instruction (Wei et al., 2022; Wang et al., 2023; Singhal et al., 2025), instantiated as brief directives like "*Solve it in a step by step fashion*" and then state the final answer explicitly. We do not employ any prompt optimization to avoid introducing confounding factors that could obscure the core CoT effect.

**Model and inference setting.** Details can be found in Appendix A.1.

## 4 Experiment and Results

To assess the practical impact of CoT in clinical contexts, we first investigate *what* the overall impact of CoT is on performance. Then, we analyze *how* CoT behaves by examining the relationship between reasoning length and performance as well as the alignment of generated content with medical concepts. Finally, we investigate *why* CoT fails via an error taxonomy supported by LLM-as-a-Judge assessments and clinical expert validation.

### 4.1 CoT Undermines LLM Performance in Clinical Text Tasks

**Setup.** For each model $M$, we compute $\overline{\Delta Score}_M$ and $\overline{\Delta Score(\%)}_M$ to summarize the overall CoT effect across tasks (See 3.3). We report macro-averages at the model level and further stratify by model capability (quartiles based on $\overline{S}_m^{\text{zs}}$).

**Main result and findings.** As shown in Figure 1, most LLMs (82 of 95, 86.3%) exhibit a performance reduction under CoT prompting. Full model-wise scores are provided in Appendix Table S1. Specially, even the best-performing LLMs decline under CoT: *Gemini-2.5-flash* $(-1.6, -3.5\%)$, *Deepseek-R1* $(-2.1, -4.9\%)$, and *GPT-4o* $(-3.5, -8.0\%)$. Among the 13 models with improvements, gains are small $(0.1–2.2$ points) and 10/13 are below 1 point, indicating marginal improvements.

To verify the robustness of this trend, we hand-crafted four semantically equivalent prompt pairs for each task $(p^{\text{zs}}$ and $p^{\text{cot}})$ and re-ran evaluations on three widely used

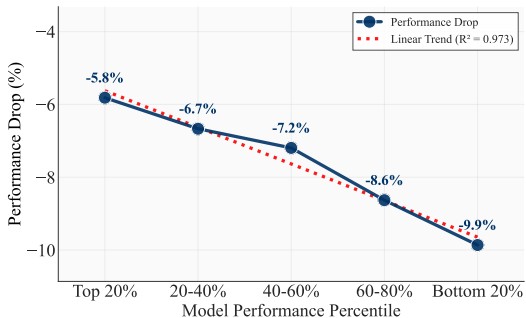

Figure 3: CoT prompting shows diminishing negative impact as model capability increases.

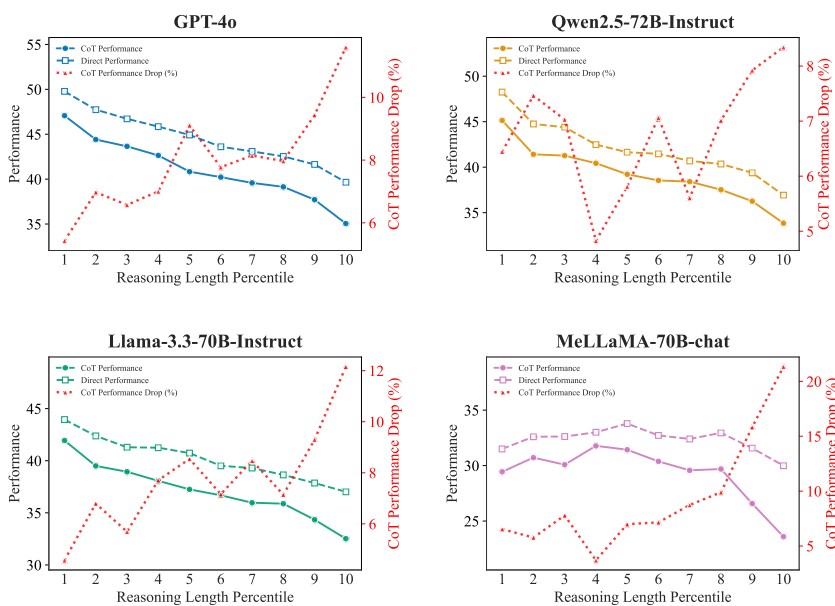

Figure 4: Performance under CoT prompting decreases as reasoning length increases.

open-source model families, *Llama-3.3-70B-Instruct*, *Mistral-Large-Instruct-2411*, *Qwen2.5-72B-Instruct* and one high-performing variant (*Athene-V2-Chat*). Across all variants, the direction of the effect remained consistent, and the magnitudes are were comparable (See Appendix Table S7), supporting the robustness of this global finding.

Grouping models by zero-shot capability reveals a graded pattern: higher-capability models are relatively more robust under CoT, whereas lower-capability models suffer larger drops. The linear relationship ($R^2 = 0.973$) implies a systematic interaction between model capability and the effectiveness of reasoning strategies. This stratification suggests that generating reliable reasoning imposes higher competence demands on the model.

## 4.2 Longer Reasoning Is Associated with Higher Performance Decline

**Setup.** We extract the *reasoning trace* $r_i$ (the text before the final answer) using the output format required by $p_t^{\text{cot}}$. The reasoning length $L(r_i)$ is measured as the word count of $r_i$; word-level counts avoid tokenizer-specific artifacts. Within each model–task pair, instances are sorted by $L(r_i)$ and partitioned into ten equal-sized bins (deciles). For each decile $d$, we compute the task score under CoT, $S_t^{\text{cot}}(d)$, and—on the *same items*—the zero-shot score, $S_t^{\text{zs}}(d)$. We then summarize the length-conditioned relative change

$$\Delta Score(d)(\%) \ = \ \frac{S_t^{\text{cot}}(d) - S_t^{\text{zs}}(d)}{S_t^{\text{zs}}(d)}.$$

Pairing by instance controls for difficulty: each decile compares CoT and zero-shot on identical inputs. We report results for four models—*GPT-4o, Qwen2.5-72B-Instruct, Llama-3.3-70B-Instruct*, and *MeLLaMA-70B-chat*, which are representative proprietary and open-source model families, and one medical variant.

**Main results and findings.** Figure 4 shows a clear negative association between reasoning length and performance for all four models. Both CoT and zero-shot scores decrease as $L(r)$ increases, reflecting that longer traces often correspond to harder instances. However, the *relative* CoT drop grows with length: $\Delta\%(d)$ becomes increasingly negative from short to long deciles. For example, *GPT-4o* exhibits a widening decline of roughly 5–15 percentage points across deciles, while *MeLLaMA-70B-chat* ranges from about 5 to 22 points. Zero-shot scores computed on the identical items are comparatively flat or decline much less, indicating that the amplified degradation is not merely a consequence of instance difficulty.

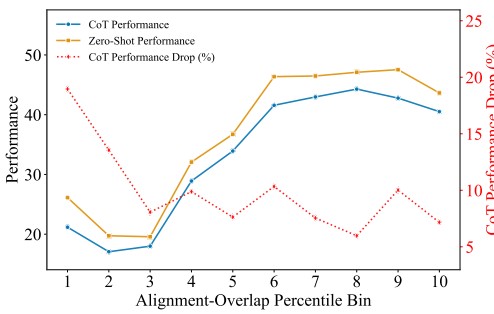 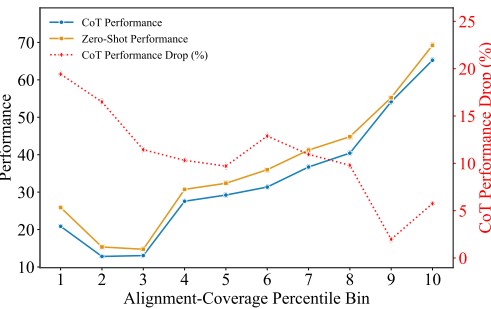

Figure 5: Impact of medical concept alignment on performance. Both Overlap and Coverage alignment metrics show systematic performance improvements across percentiles.

### 4.3 Medical Concept Alignment Mitigates CoT Degradation

**Setup.** To examine how clinically grounded content relates to CoT reliability, we extract medical concepts from both the input $x_i$ and the CoT reasoning trace $r_i$ using cTAKES (Savova et al., 2010) (version 4.0.0.1), a clinical text analysis and knowledge extraction system that extracts clinical entities in free text and maps them into Concept Unique Identifiers (CUIs). For each instance, we define the set of input concepts $C_x$ and the set of concepts mentioned in the reasoning trace $C_r$ after normalization and de-duplication. We quantify *alignment* using two metrics to reflect the concept alignment:

$$\text{Alignment-Overlap} = Jaccard(C_r, C_x) = \frac{|C_r \cap C_x|}{|C_r| \cup |C_x|}, \quad \text{Alignment-Coverage} = \frac{|C_r \cap C_x|}{|C_x|}W$$

Intuitively, Alignment-Overlap captures the extent to which the concepts *introduced by the model* overlap with those already present in the input, while Alignment-Coverage measures how much of the *input's clinical content* is covered by the reasoning trace.

Following the above length analysis protocol, we rank instances within each model–task pair by the alignment metric and partition them into ten equal-sized bins (deciles). For each decile $d$, we compute the task score under CoT, $S_t^{\text{cot}}(d)$, and—on the identical instances—the zero-shot score, $S_t^{\text{zs}}(d)$, and report the relative change (Sec. §4.2). To capture representative model behavior while ensuring sufficient CoT data, we include models whose valid-response rate under CoT exceeds 90%.

**Main results and findings.** Figure 5 summarizes the results. Both task performance and the CoT performance drop improve with stronger concept alignment. When Alignment-Overlap (left) increases from the lowest to the highest deciles, average scores rise steadily for both protocols, and the relative CoT deficit shrinks. A similar pattern holds for Alignment-Coverage (right): covering more of the input's clinical concepts is associated with higher accuracy and a smaller CoT penalty. These trends indicate that CoT degradation is attenuated when the reasoning trace remains tightly grounded in the clinical content of the input, suggesting that concept-faithful rationales are a key precondition for reliable CoT.

### 4.4 Lexical Signatures of CoT Reasoning Trace for Correct and Incorrect Instances

**Setup.** We partition the evaluation instances into two disjoint groups based on the correctness of the *final answer* under CoT: *CoT-correct* and *CoT-incorrect*. For each instance, we take the CoT reasoning trace $r_i$ and compute corpus-level word statistics after lowercasing, punctuation stripping, and lemmatization. To reduce confounding from input copying, we remove tokens that also appear in the corresponding input $x_i$. Let $V$ be the vocabulary after stopword removal. For a term $w \in V$, let $f_w^{\text{cor}}$ and $f_w^{\text{inc}}$ denote its token counts in CoT-correct and CoT-incorrect traces, with $N^{\text{cor}} = \sum_w f_w^{\text{cor}}$ and $N^{\text{inc}} = \sum_w f_w^{\text{inc}}$. We report (i) normalized frequencies $p_w^{\text{cor}} = f_w^{\text{cor}}/N^{\text{cor}}$ and $p_w^{\text{inc}} = f_w^{\text{inc}}/N^{\text{inc}}$ and (ii) differential usage

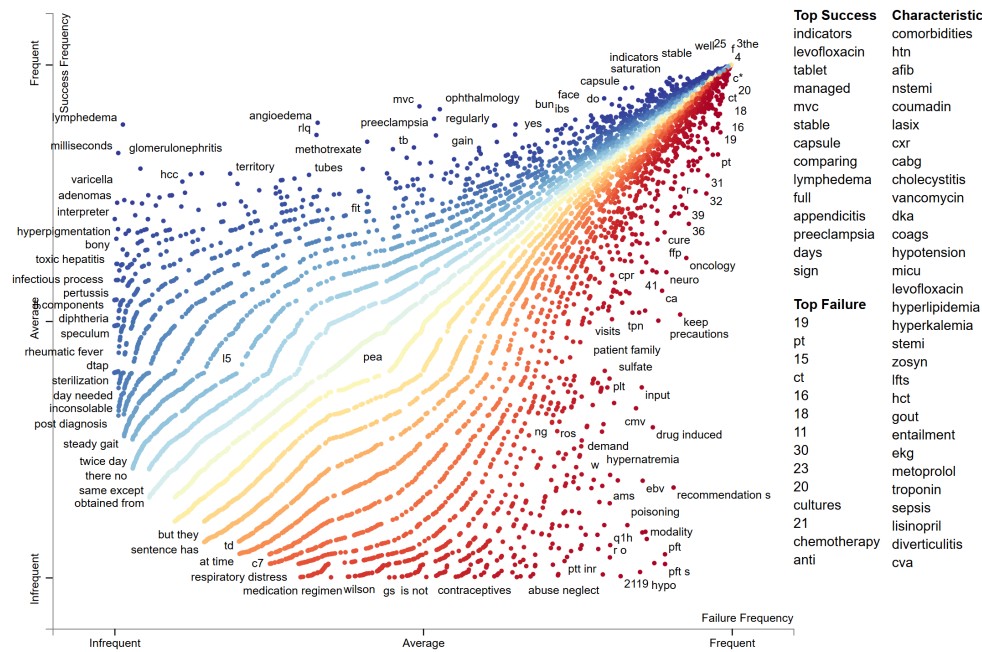

Figure 6: Lexical contrast between correct and incorrect CoT. Each point is a word; axes show normalized frequency in two groups. Color encodes density from infrequent (yellow) to frequent (blue/red). The right panels list the most characteristic words for each group.

via informative log-odds with a Dirichlet prior (Monroe et al., 2008; Kessler, 2017):

$$\delta_w = \log \frac{f_w^{\text{cor}} + \alpha_w}{N^{\text{cor}} + \alpha_0 - (f_w^{\text{cor}} + \alpha_w)} - \log \frac{f_w^{\text{inc}} + \alpha_w}{N^{\text{inc}} + \alpha_0 - (f_w^{\text{inc}} + \alpha_w)}, \quad z_w = \frac{\delta_w}{\sqrt{\sigma_w^2}},$$

where $\alpha_w$ is the background frequency scaled by $\alpha_0$ and $\sigma_w^2$ is the approximate variance from Monroe et al. (2008). This prior-regularized contrast mitigates small-sample artifacts and yields a ranked list of words characteristic of each group.

**Main results and findings.** Figure 6 contrasts term frequencies between CoT-correct and CoT-incorrect traces. Terms in the right-lower region (high in CoT-incorrect, low in CoT-correct) are disproportionately associated with errors. We observe that *numbers, units, and measurement-related tokens* are overrepresented among CoT-incorrect traces (*e.g.,* pt, ct, numeric values), as are abbreviations for laboratory and physiologic assessments such as plt (platelet count) and pft (pulmonary function test). Conversely, CoT-correct traces contain a higher proportion of disease and therapy terms that are semantically aligned with the input clinical background. These lexical signatures align with our earlier analyses: tasks requiring precise handling of quantitative data and shorthand notations are more vulnerable under CoT, consistent with reports that LLMs are brittle with numeric reasoning (Akhtar et al., 2023; Khandekar et al., 2024). and domain-specific abbreviations (Rajkomar et al., 2022; Kugic et al., 2024).

## 4.5 Error Analysis: Hallucination, Omission, and Incompleteness

**Setup.** To characterize failure modes in the CoT reasoning trace $r$, we adopt an LLM-as-a-Judge protocol. Following prior studies about LLM-as-a-Judge on clinical task (Liu et al., 2024; Gu et al., 2024), we evaluate three primary dimensions:

- **Hallucination.** Claims are unsupported by the input or medical knowledge.
- **Omission.** Omission of task-relevant or clinically important facts from the input.
- **Incompleteness.** Truncated, underspecified, or insufficient to justify the answer.

(a) Distribution of error combinations

| Model | Hallucination (%) | Omission (%) | Incompleteness (%) |
|---|---|---|---|
| GPT-4o | 7.7 | 1.7 | 0.2 |
| Qwen2.5-72B | 23.1 | 1.7 | 0.5 |
| Llama-3.3-70B | 25.4 | 3.0 | 0.3 |
| Me-LLaMA-70B | 27.5 | 87.8 | 96.8 |

(b) Prevalence of hallucination, omission, and incompleteness error types

Figure 7: Error taxonomy of CoT reasoning traces using LLM-as-a-Judge. Entries are the count/percentage of errors recognized (lower is better).

We adopt *OpenAI-o3* as the judge model. The judge model was manually assessed by a clinical expert in a random subset ($n = 200$). The full prompts can be found in Appendix A.10. To minimize bias, model identities are anonymized and responses are randomly ordered during judging. We analyze three challenging EHR-based tasks—mortality prediction on MIMIC-III-Outcome (van Aken et al., 2021), disease diagnosis on MIMIC-IV-DiReCT dataset (Wang et al., 2024) and MIMIC-IV-CDM dataset (Hager et al., 2024).

**Main results and findings.** Figure 7b reports the distribution of detected error types across models. Generally, the best-performing models, *GPT-4o*, show the fewest errors; the following *Qwen2.5-72B-Instruct* and *Llama-3.3-70B-Instruct* are comparable. However, *MeLLaMA-70B-Chat*, trained primarily for medical Q&A rather than EHR cases, exhibits substantially higher omission and incompleteness rates, indicating difficulty in extracting and organizing patient-specific evidence from longitudinal notes. Overall, hallucination and omission are the predominant failure modes (Figure 7a). The expert review highlights cases where historical conditions mentioned in the input are incorrectly treated as current. Compared to expert annotation, the positive predictive value (PPV) of the judge was 97.22% for hallucination, 97.83% for omission, and 100% for incompleteness, demonstrating the reliability of the LLM-as-a-Judge.

## 5 LIMITATIONS

This study has several limitations. First, the scale of expert validation is limited due to time and budget constraints; although the audited sample supports the reliability of LLM-as-a-Judge, broader expert annotation can further refine the judging rubric and provide more key insights. Second, our goal is to investigate the effectiveness of CoT in general settings; we do not evaluate engineering interventions such as self-consistency, retrieval-augmented generation (RAG), or few-shot learning, which may improve model performance.

## 6 CONCLUSION

We conducted a large-scale, systematic investigation of LLM reasoning in clinical contexts, spanning 95 LLMs and real-world clinical text tasks, and show that reliable reasoning in clinical settings sets a higher demand than general domains, with current models insufficiently adapted to clinical understanding. Analyses on CoT reasoning trace implicate error accumulation in longer chains and insufficient grounding to clinical concepts as key mechanisms, exposing weaknesses in clinical abbreviations and numerical reasoning. An LLM-as-a-Judge framework with expert review also highlights the hallucination and information omission as dominant failure modes of CoT. Overall, these results establish a foundational evidence for when and why CoT underperforms in clinical settings. More efforts are needed to enhance LLMs' understanding of medical knowledge and clinical contexts, and then develop clinically aligned reasoning strategies that reconcile transparency with trustworthy performance.

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

# A   APPENDIX

## A.1   IMPLEMENTATION DETAILS

**Model Access and Environment**   All open-source models (except Deepseek-R1) are executed locally on NVIDIA A100 GPUs ($8\times80$ GB). Deepseek-R1 and all proprietary models are accessed through an institution-managed Azure cloud service, avoiding the potential privacy concerns and security risks.

**Inference protocol and decoding controls.**   All models are evaluated under a unified decoding configuration to ensure compute parity. Unless otherwise noted, we use deterministic decoding (temperature = 0; sampling disabled). The confidence interval is estimated via bootstrap resampling with 1,000 iterations.

## A.2   MODEL PERFORMANCE

We conducted a comprehensive comparison of 95 state-of-the-art large language models (LLMs) across 87 multilingual clinical tasks derived from the BRIDGE benchmark, encompassing nine languages and eight distinct task categories (classification, extraction, question answering, summarization, and others). The evaluated models comprised proprietary systems (*e.g.*, GPT-4o and Gemini), open-source architectures (*e.g.*, DeepSeek, Llama, and Qwen), and specialized medical domain models (*e.g.*, Me-LLaMA and Meditron).

Our findings, presented in Table S1, demonstrate that most LLMs (82 of 95, 86.3%) exhibit degraded performance under CoT inference.

Table S1: Overall Performance of LLMs Across Different Inference Strategies

| Model | Model Size | Model Domain | Model Percentile | Zero-shot Score | CoT Score | Δ Score (CoT - ZS) | Δ (%) (CoT - ZS) |
|---|---|---|---|---|---|---|---|
| Gemini-2.5-flash | / | General | 80-100% | 44.8 [44.1,45.6] | 43.3 [42.5,44.1] | −1.6 | −3.5 |
| Deepseek-R1 | 671 | General | 80-100% | 44.2 [43.5,45.0] | 42.1 [41.3,42.9] | −2.1 | −4.9 |
| GPT-4o-0806 | / | General | 80-100% | 44.2 [43.4,45.0] | 40.7 [39.9,41.4] | −3.5 | −8.0 |
| Qwen3-Next-80B-A3B-Thinking | 80 | General | 80-100% | 43.9 [43.1,44.6] | 42.9 [42.1,43.7] | −1.0 | −2.2 |
| Gemini-1.5-pro-002 | / | General | 80-100% | 43.8 [43.1,44.6] | 40.5 [39.7,41.3] | −3.3 | −7.6 |
| Gemini-2.0-flash-001 | / | General | 80-100% | 43.0 [42.2,43.8] | 42.0 [41.2,42.8] | −1.1 | −2.4 |
| Mistral-Large-Instruct-2411 | 123 | General | 80-100% | 42.3 [41.5,43.1] | 38.9 [38.1,39.7] | −3.4 | −8.0 |
| Qwen3-30B-A3B-Thinking-2507 | 30 | General | 80-100% | 41.8 [41.0,42.6] | 41.4 [40.6,42.2] | −0.4 | −0.9 |
| Athene-V2-Chat | 72 | General | 80-100% | 41.7 [40.9,42.5] | 39.3 [38.6,40.1] | −2.4 | −5.6 |
| Qwen3-235B-A22B-Thinking | 235 | General | 80-100% | 41.6 [40.8,42.5] | 40.1 [39.4,40.9] | −1.5 | −3.6 |
| Qwen2.5-72B-Instruct | 72 | General | 80-100% | 41.6 [40.8,42.4] | 38.9 [38.1,39.7] | −2.8 | −6.6 |
| Qwen3-32B-Thinking | 32 | General | 80-100% | 41.0 [40.3,41.8] | 34.5 [33.7,35.3] | −6.6 | −16.0 |
| HuatuoGPT-o1-72B | 72 | Medical | 80-100% | 41.0 [40.2,41.8] | 38.1 [37.4,38.9] | −2.9 | −7.0 |
| Qwen3-30B-A3B-Thinking | 30 | General | 80-100% | 40.9 [40.1,41.8] | 39.4 [38.6,40.1] | −1.6 | −3.9 |
| medgemma-27b-it | 27 | Medical | 80-100% | 40.8 [40.0,41.6] | 38.2 [37.5,38.9] | −2.6 | −6.4 |
| Qwen3-14B-Thinking | 14 | General | 80-100% | 40.2 [39.4,41.0] | 37.1 [36.3,37.9] | −3.1 | −7.7 |
| Qwen3-4B-Thinking-2507 | 4 | General | 80-100% | 40.0 [39.2,40.8] | 38.8 [38.0,39.7] | −1.2 | −3.0 |
| Qwen3-8B-Thinking | 8 | General | 80-100% | 40.0 [39.2,40.8] | 37.1 [36.2,37.9] | −2.9 | −7.3 |
| gemma-3-27b-it | 27 | General | 80-100% | 39.9 [39.1,40.7] | 37.5 [36.8,38.3] | −2.4 | −5.9 |
| Qwen2.5-32B-Instruct | 32 | General | 60-80% | 39.9 [39.1,40.7] | 37.7 [36.9,38.5] | −2.2 | −5.6 |
| Llama-3.3-70B-Instruct | 70 | General | 60-80% | 39.9 [39.1,40.7] | 36.8 [36.1,37.6] | −3.0 | −7.6 |
| Qwen3-Next-80B-A3B-Instruct | 80 | General | 60-80% | 39.8 [39.0,40.6] | 40.5 [39.7,41.3] | 0.7 | 1.7 |
| Deepseek-R1-Distill-Llama-70B | 70 | General | 60-80% | 39.8 [39.0,40.6] | 38.9 [38.2,39.7] | −0.8 | −2.1 |
| Deepseek-R1-Distill-Qwen-32B | 32 | General | 60-80% | 39.7 [39.0,40.5] | 38.7 [37.9,39.5] | −1.0 | −2.6 |
| Mistral-Small-3.1-24B-Instruct-2503 | 24 | General | 60-80% | 39.7 [38.9,40.6] | 36.2 [35.4,37.0] | −3.5 | −8.8 |
| QWQ-32B | 32 | General | 60-80% | 39.4 [38.6,40.2] | 37.0 [36.2,37.8] | −2.3 | −6.0 |
| Qwen3-32B-Non-Thinking | 32 | General | 60-80% | 39.3 [38.5,40.1] | 32.8 [32.0,33.6] | −6.5 | −16.5 |
| QwenLong-L1-32B | 32 | General | 60-80% | 39.2 [38.4,40.1] | 36.7 [35.9,37.5] | −2.5 | −6.5 |
| Qwen3-235B-A22B-Non-Thinking | 235 | General | 60-80% | 39.2 [38.4,40.0] | 38.0 [37.2,38.8] | −1.2 | −3.1 |
| Llama-3.1-70B-Instruct | 70 | General | 60-80% | 39.1 [38.3,39.9] | 35.1 [34.3,35.9] | −4.0 | −10.2 |
| Qwen3-4B-Thinking | 4 | General | 60-80% | 38.5 [37.7,39.3] | 37.0 [36.2,37.8] | −1.5 | −4.0 |
| gemma-2-27b-it | 27 | General | 60-80% | 38.2 [37.4,39.0] | 34.2 [33.4,35.0] | −4.0 | −10.4 |
| Baichuan-M2-32B | 32 | Medical | 60-80% | 38.0 [37.0,39.0] | 38.3 [37.4,39.2] | 0.3 | 0.7 |
| Magistral-Small-2506 | 24 | General | 60-80% | 37.6 [36.8,38.4] | 34.2 [33.3,35.0] | −3.4 | −9.1 |
| Mistral-Small-24B-Instruct-2501 | 24 | General | 60-80% | 37.5 [36.7,38.4] | 31.6 [30.8,32.4] | −5.9 | −15.8 |
| gemma-3-12b-it | 12 | General | 60-80% | 37.3 [36.6,38.1] | 35.4 [34.6,36.1] | −1.9 | −5.2 |
| gpt-oss-120b | 120 | General | 60-80% | 37.2 [36.4,38.1] | 32.1 [31.4,32.9] | −5.1 | −13.8 |
| Deepseek-R1-0528-Qwen3-8B | 8 | General | 60-80% | 36.9 [36.1,37.8] | 36.3 [35.5,37.1] | −0.7 | −1.8 |
| Qwen3-30B-A3B-Instruct-2507 | 30 | General | 40-60% | 36.9 [36.1,37.6] | 37.5 [36.7,38.3] | 0.6 | 1.7 |
| Qwen3-14B-Non-Thinking | 14 | General | 40-60% | 36.8 [36.0,37.6] | 34.6 [33.8,35.4] | −2.2 | −6.0 |
| Qwen3-30B-A3B-Non-Thinking | 30 | General | 40-60% | 36.2 [35.4,36.9] | 36.9 [36.1,37.7] | 0.7 | 1.9 |
| Phi-4 | 14 | General | 40-60% | 36.1 [35.4,36.9] | 32.6 [31.8,33.4] | −3.5 | −9.8 |
| Baichuan-M1-14B-Instruct | 14 | Medical | 40-60% | 36.1 [35.3,36.9] | 34.4 [33.6,35.2] | −1.7 | −4.8 |
| gpt-35-turbo-0125 | / | General | 40-60% | 35.3 [34.5,36.1] | 31.6 [30.8,32.4] | −3.7 | −10.4 |
| Mistral-Small-Instruct-2409 | 22 | General | 40-60% | 35.2 [34.4,36.0] | 31.2 [30.4,32.0] | −4.0 | −11.4 |

Table S1 – continued from previous page

| Model | Model Size | Model Domain | Model Percentile | Zero-shot Score | CoT Score | Δ Score (CoT - ZS) | Δ (%) (CoT - ZS) |
|---|---|---|---|---|---|---|---|
| Llama-4-Scout-17B-16E-Instruct | 109 | General | 40-60% | 35.1 [34.4,35.9] | 29.4 [28.7,30.1] | −5.7 | −16.3 |
| gemma-2-9b-it | 9 | General | 40-60% | 35.1 [34.3,35.9] | 29.9 [29.2,30.7] | −5.1 | −14.6 |
| Deepseek-R1-Distill-Qwen-14B | 14 | General | 40-60% | 34.3 [33.5,35.1] | 34.8 [34.0,35.6] | 0.5 | 1.5 |
| Qwen3-4B-Instruct-2507 | 4 | General | 40-60% | 34.1 [33.3,34.8] | 36.2 [35.4,37.0] | 2.2 | 6.3 |
| HuatuoGPT-o1-70B | 70 | Medical | 40-60% | 33.9 [33.2,34.6] | 32.3 [31.6,33.1] | −1.5 | −4.6 |
| Qwen3-8B-Non-Thinking | 8 | General | 40-60% | 33.9 [33.1,34.7] | 33.3 [32.5,34.1] | −0.6 | −1.6 |
| Llama-3-70B-UltraMedical | 70 | Medical | 40-60% | 33.4 [32.6,34.2] | 29.4 [28.7,30.2] | −4.0 | −11.9 |
| Qwen3-4B-Non-Thinking | 4 | General | 40-60% | 33.3 [32.5,34.1] | 33.2 [32.4,34.0] | −0.1 | −0.3 |
| Llama3-OpenBioLLM-70B | 70 | Medical | 40-60% | 33.0 [32.2,33.8] | 28.8 [28.0,29.5] | −4.2 | −12.8 |
| Llama-3.1-Nemotron-70B-Instruct-HF | 70 | General | 40-60% | 32.7 [32.0,33.5] | 24.1 [23.5,24.7] | −8.7 | −26.4 |
| MeLLaMA-70B-chat | 70 | Medical | 40-60% | 32.3 [31.5,33.1] | 29.2 [28.4,30.1] | −3.0 | −9.3 |
| Yi-1.5-34B-Chat-16K | 34 | General | 40-60% | 32.1 [31.4,32.9] | 29.6 [28.8,30.3] | −2.5 | −7.9 |
| QwQ-32B-Preview | 32 | General | 20-40% | 31.7 [31.0,32.5] | 23.3 [22.5,24.1] | −8.4 | −26.6 |
| Qwen2.5-7B-Instruct | 7 | General | 20-40% | 31.3 [30.5,32.1] | 30.2 [29.5,31.0] | −1.1 | −3.4 |
| Ministral-8B-Instruct-2410 | 8 | General | 20-40% | 30.4 [29.5,31.2] | 25.9 [25.1,26.7] | −4.5 | −14.7 |
| HuatuoGPT-o1-7B | 7 | Medical | 20-40% | 29.6 [28.8,30.4] | 26.7 [26.0,27.4] | −2.9 | −9.8 |
| Phi-3.5-MoE-instruct | 42 | General | 20-40% | 29.5 [28.8,30.3] | 25.3 [24.5,26.0] | −4.3 | −14.5 |
| medgemma-4b-it | 4 | Medical | 20-40% | 29.4 [28.3,30.5] | 28.5 [27.6,29.3] | −0.9 | −3.1 |
| gpt-oss-20b | 20 | General | 20-40% | 29.0 [28.3,29.8] | 24.9 [24.1,25.6] | −4.2 | −14.4 |
| Llama-3.1-8B-Instruct | 8 | General | 20-40% | 29.0 [28.2,29.8] | 29.4 [28.6,30.2] | 0.4 | 1.4 |
| Yi-1.5-9B-Chat-16K | 9 | General | 20-40% | 28.8 [28.0,29.6] | 25.4 [24.6,26.2] | −3.4 | −11.9 |
| gemma-3-4b-it | 4 | General | 20-40% | 28.6 [27.8,29.4] | 28.2 [27.4,29.0] | −0.4 | −1.3 |
| Deepseek-R1-Distill-Llama-8B | 8 | General | 20-40% | 28.5 [27.7,29.2] | 27.3 [26.6,28.1] | −1.1 | −4.0 |
| Qwen2.5-3B-Instruct | 3 | General | 20-40% | 26.6 [25.8,27.3] | 25.4 [24.7,26.2] | −1.1 | −4.3 |
| Qwen3-1.7B-Thinking | 1.7 | General | 20-40% | 26.3 [25.5,27.0] | 27.7 [27.0,28.4] | 1.4 | 5.4 |
| HuatuoGPT-o1-8B | 8 | Medical | 20-40% | 25.9 [25.1,26.6] | 22.5 [21.8,23.2] | −3.4 | −13.0 |
| Phi-3.5-mini-instruct | 4 | General | 20-40% | 25.4 [24.7,26.1] | 23.9 [23.2,24.6] | −1.5 | −5.9 |
| Deepseek-R1-Distill-Qwen-7B | 7 | General | 20-40% | 25.3 [24.5,26.0] | 23.9 [23.1,24.7] | −1.4 | −5.5 |
| Llama-2-70b-chat | 70 | General | 20-40% | 25.2 [24.4,25.9] | 19.0 [18.3,19.7] | −6.1 | −24.4 |
| Phi-4-mini-instruct | 4 | General | 20-40% | 24.5 [23.8,25.3] | 22.5 [21.8,23.2] | −2.0 | −8.3 |
| Llama-3.2-3B-Instruct | 3 | General | 20-40% | 22.9 [22.1,23.7] | 21.6 [20.9,22.3] | −1.3 | −5.7 |
| Qwen2.5-1.5B-Instruct | 1.5 | General | 0-20% | 22.2 [21.4,22.9] | 19.5 [18.7,20.2] | −2.7 | −12.1 |
| Qwen3-1.7B-Non-Thinking | 1.7 | General | 0-20% | 22.0 [21.2,22.7] | 22.8 [22.0,23.5] | 0.8 | 3.7 |
| Phi-4-mini-reasoning | 4 | General | 0-20% | 21.3 [20.6,22.0] | 19.8 [19.1,20.5] | −1.5 | −7.1 |
| Llama-2-13b-chat | 13 | General | 0-20% | 20.9 [20.2,21.7] | 16.2 [15.5,16.9] | −4.7 | −22.4 |
| MeLLaMA-13B-chat | 13 | Medical | 0-20% | 20.8 [20.0,21.5] | 20.3 [19.5,21.0] | −0.5 | −2.4 |
| BioMistral-7B | 7 | Medical | 0-20% | 20.4 [19.7,21.2] | 10.8 [10.2,11.5] | −9.6 | −46.9 |
| Qwen3-0.6B-Thinking | 0.6 | General | 0-20% | 20.4 [19.6,21.1] | 18.9 [18.2,19.7] | −1.4 | −7.1 |
| MMed-Llama-3-8B | 8 | Medical | 0-20% | 20.4 [19.7,21.1] | 16.2 [15.4,16.9] | −4.2 | −20.6 |
| Llama-3.1-8B-UltraMedical | 8 | Medical | 0-20% | 20.2 [19.5,20.9] | 18.3 [17.7,19.0] | −1.8 | −9.0 |
| OpenThinker3-7B | 7 | General | 0-20% | 19.8 [19.0,20.5] | 18.4 [17.7,19.1] | −1.4 | −7.2 |
| MedReason-8B | 8 | Medical | 0-20% | 18.2 [17.4,18.9] | 18.3 [17.6,19.0] | 0.1 | 0.7 |
| Llama-2-7b-chat | 7 | General | 0-20% | 16.5 [15.8,17.2] | 13.7 [13.0,14.3] | −2.8 | −17.1 |
| gemma-3-1b-it | 1 | General | 0-20% | 15.7 [15.0,16.4] | 13.5 [12.8,14.2] | −2.2 | −14.0 |
| meditron-70b | 70 | Medical | 0-20% | 15.7 [15.0,16.4] | 13.2 [12.5,13.9] | −2.5 | −16.0 |
| Qwen3-0.6B-Non-Thinking | 0.6 | General | 0-20% | 15.2 [14.5,15.9] | 16.6 [15.9,17.3] | 1.4 | 9.1 |
| Deepseek-R1-Distill-Qwen-1.5B | 1.5 | General | 0-20% | 14.3 [13.6,15.0] | 13.4 [12.7,14.1] | −0.8 | −5.9 |
| Llama3-OpenBioLLM-8B | 8 | Medical | 0-20% | 14.2 [13.5,14.9] | 13.3 [12.6,14.0] | −0.9 | −6.4 |
| Llama-3.2-1B-Instruct | 1 | General | 0-20% | 12.7 [12.1,13.4] | 11.9 [11.2,12.5] | −0.9 | −6.7 |
| Phi-4-reasoning-plus | 14 | General | 0-20% | 10.6 [10.0,11.2] | 10.8 [10.2,11.4] | 0.3 | 2.5 |
| Phi-4-reasoning | 14 | General | 0-20% | 10.0 [9.4,10.6] | 10.1 [9.5,10.7] | 0.1 | 0.9 |
| meditron-7b | 7 | Medical | 0-20% | 9.5 [8.9,10.2] | 9.5 [8.9,10.1] | 0.0 | 0.0 |

## A.3 MODEL PERFORMANCE PERCENTILE

We defined a model's basic capability as its macro-average zero-shot performance across all BRIDGE tasks. To examine whether model capability influences susceptibility to CoT-induced performance degradation, we stratified the 95 models into five equal percentile tiers (19 models each) to analyze this relationship systematically.

As shown in Table S2, higher-performing models are more resilient to CoT inference, with top-tier models showing smaller performance drops than lower-performing counterparts.

Table S2: Performance Drop of CoT Prompting Across LLM Strength Tiers

| Model Performance Tier | Δ Score (%) CoT | Observation |
|---|---|---|
| Top 20% | -5.82 | Smallest gap |
| 20%–40% | -6.67 | Moderate |
| 40%–60% | -7.19 | Moderate |
| 60%–80% | -8.63 | Large |
| Bottom 20% | -9.86 | Largest gap |

## A.4 EVALUATION OF COT COMBINING FEW-SHOT

**Methods.** LLMs can leverage the capability of in-context learning and few-shot prompting to understand task definitions and expected outputs better.Brown et al. (2020) In this setting, the examples with explicit chain-of-thought (CoT) rationales could also further guide models toward step-by-step clinical reasoning.Nachane et al. (2024)

To examine the practical effect of CoT combining few-shot prompting on clinical text tasks, we conduct a few-shot study on 87 clinical text tasks across LLMs. For each task, we randomly select five additional examples from outside the testing samples and construct two types of in-context demonstrations:

- QA (Few-shot): Each example contains only the question and its correct answer, serving as standard few-shot demonstrations.
- CoT (Few-shot): Each example contains the question, a detailed CoT reasoning process of question-solving, and the correct answer.

For the CoT-augmented examples, we generate the reasoning traces using Qwen3-Next-80B-A3B-Thinking, which is the second-best model among all 95 LLMs and the best-performing open-source model under CoT prompting in our evaluation. As the higher cost of time and computing, we focus on seven representative models (five general-purpose and two medical LLMs) and evaluate them on all 87 clinical text tasks under four conditions: QA (zero-shot ), CoT (zero-shot ), QA (few-shot), and CoT (few-shot). As in the main experiments, we report the overall score (averaged across tasks) for each setting.

Table S3: Model performance under zero-shot and few-shot. (QA=only question and correct answer; CoT=question, analysis, and correct answer; same as the following charts.)

| Model Domain | Model | Zero-shot | | | Few-Shot | | |
| --- | --- | --- | --- | --- | --- | --- | --- |
| | | QA | CoT | $\Delta$(CoT − QA) (%) | QA | CoT | $\Delta$(CoT − QA) (%) |
| General | Qwen2.5-7B-Instruct | 31.32 | 30.25 | −1.07 ( -3.42%) | 41.60 | 35.47 | −6.13 ( -14.74%) |
| General | Qwen2.5-72B-Instruct | 41.62 | 38.86 | −2.76 ( -6.63%) | 50.99 | 49.90 | −1.09 ( -2.14%) |
| General | Llama-3.1-8B-Instruct | 28.98 | 29.40 | +0.42 (+1.45%) | 43.54 | 40.99 | −2.55 ( -5.86%) |
| General | Llama-3.1-70B-Instruct | 39.09 | 35.10 | −3.99 ( -10.21%) | 50.52 | 50.23 | −0.29 ( -0.57%) |
| General | Athene-V2-Chat | 41.69 | 39.34 | −2.35 ( -5.64%) | 50.68 | 50.38 | −0.30 ( -0.59%) |
| Medical | MedGemma-27B-it | 40.79 | 38.20 | −2.59 ( -6.35%) | 51.97 | 50.79 | −1.18 ( -2.27%) |
| Medical | HuatuoGPT-o1-72B | 41.01 | 38.15 | −2.86 ( -6.97%) | 50.21 | 46.62 | −3.59 ( -7.15%) |

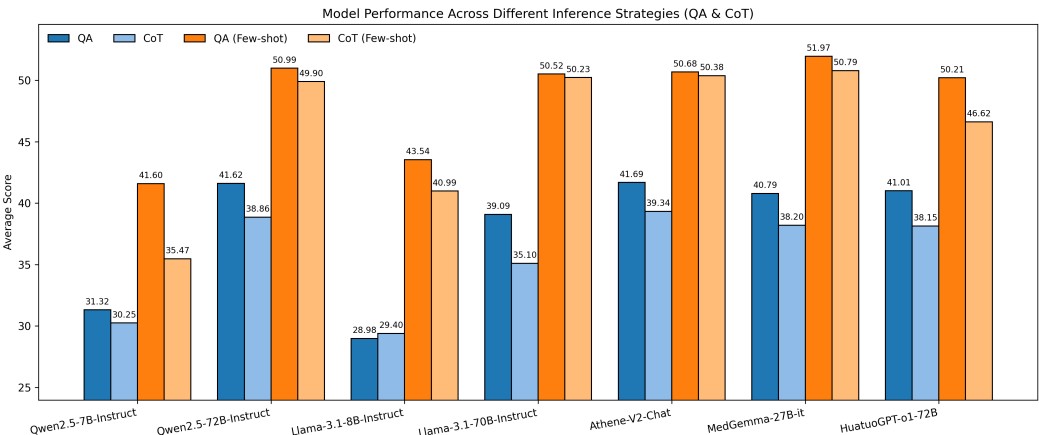

Figure S1: Performance comparison across different inference strategies.

**Results.** Table S3 and figure S1 summarize the results. For each model, we report: (i) zero-shot QA and zero-shot CoT scores; (ii) the absolute and relative score change $\Delta score =$

$(\text{CoT} - \text{QA})$ and $\Delta score(\%) = (\text{CoT} - \text{QA})/\text{QA}(\%)$; and (iii) the corresponding scores under few-shot QA and few-shot CoT, along with the relative change under few-shot prompting.

Across all seven models, few-shot QA substantially improves performance compared with zero-shot QA (e.g., Qwen2.5-72B-Instruct from 41.62 to 50.99, MedGemma-27B-it from 40.79 to 51.97). In contrast, adding CoT on top of few-shot QA (*Few-shot CoT*) does not yield further gains and typically leads to a small performance drop relative to few-shot QA (e.g., Qwen2.5-7B-Instruct: $-6.13$ points, Llama-3.1-8B-Instruct: $-2.55$ points).

**Discussion.**   Based on the evaluation results, we can observe that: First, few-shot prompting significantly improves performance over zero-shot across LLMs on clinical text tasks, confirming the effectiveness of in-context learning for LLMs. Second, augmenting few-shot exemplars with CoT does *not* provide additional benefits and, in most cases, still slightly degrades performance relative to few-shot QA, reinforcing our main finding that CoT is not reliably helpful for clinical text tasks. Third, for 4 out of 7 LLMs (Llama-3.1-70B-Instruct, Qwen2.5-72B-Instruct, Athene-V2-Chat, and MedGemma-27B-it), the magnitude of CoT-induced degradation is smaller under few-shot than under zero-shot, indicating that in-context demonstrations may partially mitigate the negative impact of CoT while preserving most of the gains from few-shot learning. Overall, few-shot prompting improves robustness but does not overturn the main conclusion that CoT prompting is often suboptimal for real-world clinical text tasks.

### A.5   EVALUATION OF CoT INTEGRATING SELF-CONSISTENCY

**Methods.**   Self-consistency is a simple ensembling strategy that samples multiple reasoning chains and aggregates their predictions, which may stabilize CoT behavior. Wang et al. (2023) To efficiently study its impact in our setting, we focus on a subset of 27 classification tasks from BRIDGE, including text classification, natural language inference (NLI), and semantic similarity.

For five representative models (four general-purpose and one medical LLM), we compare four decoding strategies:

- QA: Standard QA without CoT, using greedy decoding.
- CoT: CoT prompting with a single reasoning chain, using greedy decoding.
- QA (Self-Consistency): QA with self-consistency: we sample 5 independent predictions and aggregate the final label by majority vote.
- QA (Self-Consistency): CoT with self-consistency: we sample 5 independent CoT reasoning chains and aggregate the final label by majority vote.

We use the same task instructions and QA/CoT prompts as in the main experiments. For greedy decoding, we set temperature$= 0$ and disable sampling. For self-consistency, we enable sampling, adopting each model's officially recommended sampling configuration when available; otherwise, we use temperature=1 and top_p=1. In all cases, we report the average score across the 27 classification tasks.

**Results.**   Table S4 summarizes the results. For each model, we report QA and CoT scores under greedy decoding, together with the absolute and relative change $\Delta(\text{CoT} - \text{QA})$ in the zero-shot setting, and the corresponding scores under self-consistency with 5 samples, along with the relative change under self-consistency. Figure S2 visualizes the average performance of these models under greedy decoding and self-consistency. Overall, self-consistency (with 5 samples) tends to improve or maintain QA performance, and narrows the performance gap between QA and CoT relative to the greedy setting.

**Discussion.**   The results of self-consistency reveal that: first, as expected from ensembling, self-consistency generally improves or slightly boosts QA performance compared with single-sample greedy decoding across the evaluated models. Second, for most models, CoT with

Table S4: Model performance under greedy decoding and self-consistency with sampling.

| Model Domain | Model | Zero-shot | | | Self-Consistency | | |
|---|---|---|---|---|---|---|---|
| | | QA | CoT | $\Delta$(CoT − QA) (%) | QA | CoT | $\Delta$(CoT − QA) (%) |
| General | Qwen2.5-72B-Instruct | 70.49 | 68.58 | -1.91 (-2.71%) | 72.31 | 71.54 | -0.77 (-1.06%) |
| General | Llama-3.1-70B-Instruct | 68.12 | 66.61 | -1.51 (-2.22%) | 69.59 | 69.62 | 0.03 ( 0.04%) |
| General | Athene-V2-Chat | 70.51 | 69.05 | -1.46 (-2.07%) | 72.45 | 72.06 | -0.39 (-0.54%) |
| General | Qwen3-Next-80B-A3B-Thinking | 73.11 | 73.04 | -0.07 (-0.10%) | 73.77 | 73.19 | -0.58 (-0.79%) |
| Medical | HuatuoGPT-o1-72B | 72.32 | 69.56 | -2.76 (-3.82%) | 72.07 | 71.84 | -0.23 (-0.32%) |

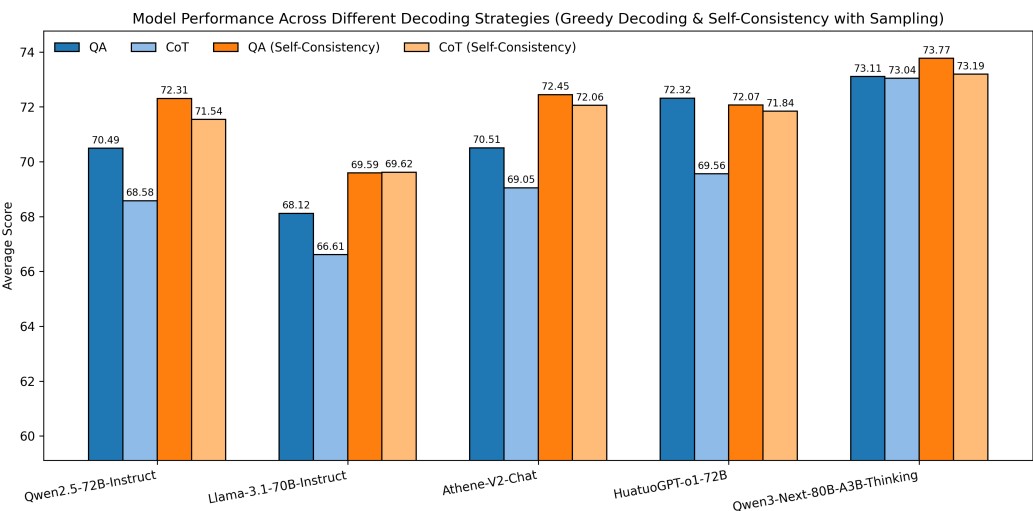

Figure S2: Performance comparison across different decoding strategies (greedy decoding vs. self-consistency with sampling) on 27 classification-style tasks.

self-consistency does *not* surpass QA with self-consistency, but it substantially reduces the performance gap between QA and CoT that we observe under greedy decoding, making CoT behavior more stable than naive single-chain CoT. For example, Qwen2.5-72B-Instruct and HuatuoGPT-o1-72B exhibit smaller negative deltas under 5-sample self-consistency than in the greedy setting, and Llama-3.1-70B-Instruct shows a slight advantage for CoT under self-consistency.

However, in practice, self-consistency introduces non-trivial computational overhead, as it requires multiple generations per query, especially for CoT where each sample is a long reasoning trace. In real-world clinical workflows, this also implies additional effort to inspect or select among multiple reasoning chains if explanations are surfaced to end users. Taken together, self-consistency can partially mitigate CoT-induced degradation and improve robustness, but CoT with self-consistency is not reliably beneficial for clinical text classification tasks and comes with additional computational and operational costs.

### A.6 PERFORMANCE ANALYSIS ACROSS TASK TYPES

**Methods.** To explore how CoT effects vary across different types of clinical NLP tasks, we further conduct the performance analysis across eight categories of 87 clinical text tasks: text classification, normalization and coding, semantic similarity, natural language inference (NLI), event extraction, named entity recognition (NER), question answering (QA), and summarization. For each task type, the average performance of several representative LLMs was calculated, including three general-purpose models (GPT-4o, Qwen2.5-72B-Instruct, and Llama-3.3-70B-Instruct) and one medical model (MedGemma-27B-it).

For each model and task type, we report the mean score under standard QA and under CoT prompting, together with the relative change $\Delta$(CoT − QA)/QA (%). We also report an *Overall* row that averages the relative changes across the four models for each task type.

Table S5: Subgroup performance of LLMs across eight task types. Each entry shows QA / CoT and the relative change. The overall row reports the average relative change across all models for each type. Bold values indicate cases where CoT improves performance.

| Domain | Model | Text Clf. | Norm./Coding | Similarity | NLI | Event Ext. | NER | QA | Summ. |
|---|---|---|---|---|---|---|---|---|---|
| General | GPT-4o | 69.16 / 68.81 (-0.50%) | 8.59 / 7.19 (-16.31%) | 47.90 / 41.85 (-12.63%) | 87.37 / 85.62 (-2.00%) | 29.05 / 23.26 (-19.93%) | 45.27 / 39.23 (-13.36%) | 16.82 / 14.41 (-14.34%) | 33.55 / 29.25 (-12.81%) |
| General | Qwen2.5-72B-Instruct | 68.52 / 67.27 (-1.83%) | 4.30 / **5.90** (+37.16%) | 47.60 / 44.56 (-6.38%) | 83.44 / 80.04 (-4.08%) | 27.71 / 21.54 (-22.27%) | 38.00 / 36.39 (-4.24%) | 17.76 / 15.64 (-11.92%) | 33.83 / 26.53 (-21.58%) |
| General | Llama-3.3-70B-Instruct | 65.27 / 65.13 (-0.21%) | 5.34 / **6.20** (+16.07%) | 41.86 / 36.02 (-13.95%) | 83.81 / 83.03 (-0.93%) | 24.23 / 18.18 (-24.96%) | 38.47 / 34.28 (-10.89%) | 16.71 / 15.75 (-5.74%) | 32.82 / 24.47 (-25.45%) |
| Medical | MedGemma-27B-it | 66.88 / 65.56 (-1.98%) | 3.31 / **4.12** (+24.49%) | 44.65 / 39.56 (-11.39%) | 82.19 / **85.38** (+3.89%) | 24.44 / 18.85 (-22.86%) | 40.98 / **41.78** (+1.95%) | 16.70 / 8.89 (-46.76%) | 34.13 / 21.40 (-37.29%) |
| – | Overall | -0.76 (-1.13%) | **0.47 (+8.66%)** | -5.00 (-10.99%) | -0.68 (-0.81%) | -5.90 (-22.38%) | -2.76 (-6.79%) | -3.32 (-19.56%) | -8.17 (-24.33%) |

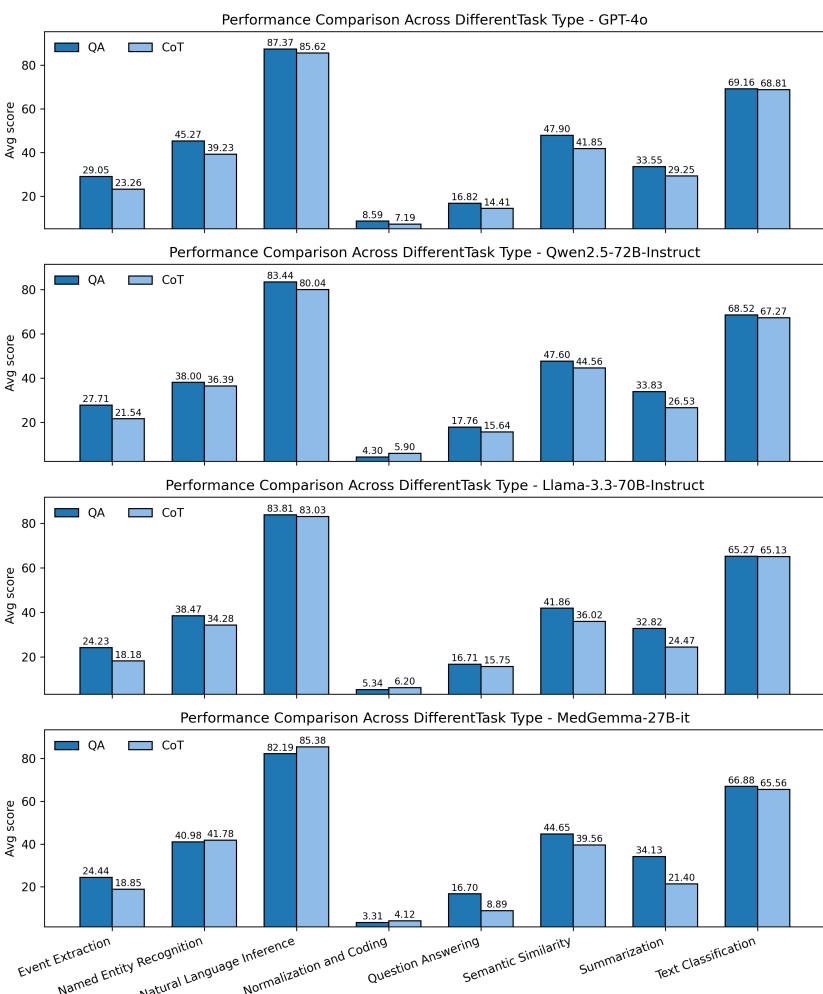

Figure S3: Performance comparison across different task types.

**Results.** Table S5 summarizes the average performance of LLMs across the eight task types. For visualization, we additionally provide a bar-plot comparison of the average relative changes across task types (see Figure S3).

**Discussion.** Overall, the task-type breakdown shows that CoT tends to degrade performance across most clinical NLP tasks. The largest average relative drops appear in event extraction (about −22%), summarization (about −24%), and question answering (about −20%). The results suggest that long, free-form CoT traces can be particularly harmful when the model must integrate many clinical details into a coherent decision.

For text classification and natural language inference, the average effect of CoT is smaller in magnitude (around −1 to −3%) but still generally negative across models, indicating that even relatively simple decision tasks do not consistently benefit from vanilla CoT in clinical text. Normalization and coding is the only task type where CoT yields a consistent average improvement (about +8.7%), albeit from relatively low absolute scores, suggesting that structured mapping problems to controlled vocabularies may occasionally profit from step-by-step reasoning.

Information extraction tasks provide a more nuanced picture. Clinical NER and event extraction are often highly context-dependent, involving temporality (current vs. historical conditions), negation, uncertainty, and cross-sentence linkage, so in principle CoT explanations could help users understand *why* certain spans are extracted. However, CoT does not impair performance on NER more than on other task types (average drops of about −7%), and event extraction largely follows the same degradation pattern as other information-rich generation tasks such as summarization and QA.

Generally, the CoT is hard to improve LLMs' capability in real-world clinical text understanding, and its negative impact is widespread across classification, inference, QA, summarization, and extraction, with normalization/coding being a notable but restricted exception.

### A.7 PERFORMANCE ANALYSIS ACROSS LANGUAGES

**Methods.** To investigate how CoT effects vary across different languages, we perform a language-wise subgroup analysis. We group all tasks by their language and, for each of the nine languages: English (En), Portuguese (Pt), Spanish (ES), Chinese (Zh), Japanese (Jp), Russian, (Ru) French (Fr), German (De), and Norwegian (No), compute the average performance of four representative LLMs: three general-purpose models (GPT-4o, Qwen2.5-72B-Instruct, Llama-3.3-70B-Instruct) and one medical model (MedGemma-27B-it).

For each model and language, we report the mean score under standard QA (without CoT) and under CoT prompting, together with the relative change $\Delta(\text{CoT} - \text{QA})/\text{QA}$ (%). We also report an "*Overall*" row that averages the relative changes across the four models for each language. Because the mix of tasks and difficulty varies across languages, these language-wise averages are intended for *within-language* comparison of QA vs. CoT, rather than for direct cross-language ranking.

**Results.** Table S6 shows the average performance of the four models across the nine languages. Meanwhile, we also plot the average relative changes per language (see Figure S4), which highlights the cross-language pattern of CoT-induced degradation.

Table S6: Average performance of LLMs across nine languages under QA vs. CoT prompting. Each cell shows QA / CoT and the relative change of CoT vs. QA. The overall row reports the average relative change across all models for each language. Bold values indicate cases where CoT improves performance.

| Domain | Model | En | Pt | Es | Zh | Ja | Ru | Fr | De | No |
|---|---|---|---|---|---|---|---|---|---|---|
| General | GPT-4o | 47.19 / 43.10 (-8.68%) | 57.69 / **58.65** (+1.67%) | 25.40 / 22.82 (-10.15%) | 43.86 / 41.19 (-6.08%) | 37.40 / 30.82 (-17.61%) | 64.71 / 63.98 (-1.13%) | 48.82 / 38.71 (-20.72%) | 52.16 / 46.81 (-10.26%) | 54.15 / 53.57 (-1.07%) |
| General | Qwen2.5-72B-Instruct | 45.54 / 42.76 (-6.09%) | 61.26 / 53.66 (-12.41%) | 23.75 / 20.29 (-14.59%) | 41.71 / 39.44 (-5.43%) | 34.48 / 31.59 (-8.38%) | 59.55 / 59.18 (-0.61%) | 50.02 / 40.01 (-20.01%) | 42.09 / **44.08** (+4.73%) | 38.19 / **39.11** (+2.40%) |
| General | Llama-3.3-70B-Instruct | 44.60 / 39.30 (-11.90%) | 58.66 / 53.59 (-8.64%) | 21.83 / **21.88** (+0.22%) | 37.56 / 37.14 (-1.12%) | 30.96 / 25.78 (-16.73%) | 58.79 / **59.93** (+1.95%) | 45.56 / 29.59 (-35.06%) | 47.19 / **47.49** (+0.64%) | 36.38 / **40.03** (+10.04%) |
| Medical | MedGemma-27B-it | 44.62 / 39.18 (-12.19%) | 59.37 / 53.29 (-10.25%) | 20.84 / **21.57** (+3.48%) | 39.83 / 37.30 (-6.35%) | 34.22 / 32.94 (-3.76%) | 61.40 / **64.05** (+4.32%) | 48.26 / 41.97 (-13.03%) | 48.84 / **54.22** (+11.02%) | 41.14 / **43.28** (+5.21%) |
| – | Overall | -4.40 (-9.68%) | -4.45 (-7.51%) | -1.32 (-5.74%) | -1.97 (-4.84%) | -3.99 (-11.63%) | **0.68** (**+1.11%**) | -10.60 (-22.00%) | **0.58** (**+1.22%**) | **1.53** (**+3.61%**) |

**Discussion.** The language-wise analysis shows that CoT-induced degradation is a multilingual phenomenon. Averaged across the four representative models, CoT reduces performance in most languages, with relative drops of roughly −4% to −12% in English, Portuguese, Spanish, Chinese, and Japanese, and an even larger average decline in French (around −22%). Russian and German are close to neutral on average (about +1.1% and +1.2%), and Norwegian is the only language with a modest average improvement (about

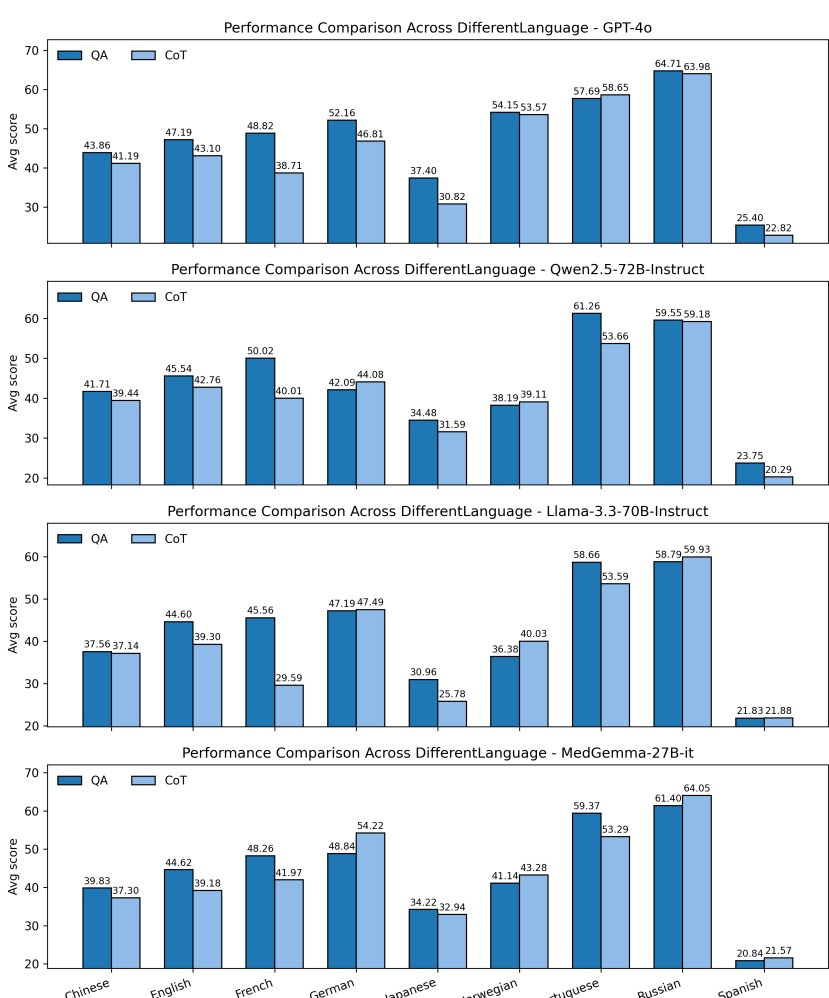

Figure S4: Performance comparison across different languages.

+3.6%). This pattern suggests that the instability of vanilla CoT on clinical text appears across multiple linguistic settings.

At the same time, the magnitude and even the direction of CoT's impact vary by both language and model. Certain language–model pairs (e.g., GPT-4o in Portuguese; Qwen2.5-72B-Instruct in German and Norwegian; MedGemma-27B-it in Spanish, Russian, German, and Norwegian) exhibit small gains under CoT, whereas others (notably Japanese and French for several general-purpose models) suffer substantial relative declines. These differences should be interpreted qualitatively rather than as strict cross-language comparisons, because the underlying task mix and difficulty differ by language.

Taken together, the language-specific analysis confirms that CoT is not reliably beneficial on real-world clinical text—holds across multiple languages, while also highlighting model- and language-dependent variability. This underscores the need to validate CoT behavior in the specific language and clinical setting of interest, rather than assuming that positive or negative effects observed in English will directly transfer to other languages.

## A.8 Case study of error analysis

Integrating error analysis supported by the LLM-as-a-judge framework and clinical expert review, we provide a case study to give a more intuitive view of how CoT can fail in practice. As the clinical dataset used in this section is regulated, we omit most details and lightly rephrase some sentences to protect privacy while preserving the medically relevant meanings.

---

**Task:** MIMIC-III Outcome.Mortality.

**Task instruction (CoT):**

> Given a patient's basic information and admission notes, predict the patient's in-hospital mortality, which means whether the patient will die during the current admission.
>
> Solve it in a step-by-step fashion, return your answer in the following format, PROVIDE DETAILED ANALYSIS BEFORE THE RESULT:
>
> Analysis:
>
> . . .
>
> Result:
>
> In-Hospital Mortality: label
>
> The optional list for "label" is ["Yes", "No"].

**Input (excerpt):**

> The clinical note includes vital signs such as: `PHYSICAL EXAM:`
> `VS: T 95.2 BP 132/43 HR 74 RR 15 O2sat 100% NRB`
> (here, "NRB" denotes a non-rebreather mask.)

**Output 1 (Llama-3.3-70B-Instruct, CoT):**

> In its step-by-step analysis, the model writes:
> *"The patient's physical exam revealed a temperature of 95.2, blood pressure of 132/43, and oxygen saturation of 100% on room air."*

**LLM-judge (Flags hallucination):**

> States that the patient had "O2 saturation 100% on room air," whereas the note specifies "O2sat 100% NRB" (non-rebreather), so the room-air claim is unsupported. Also labels BP 132/43 outright as hypotension; this is debatable and not in the input. Hence, the response includes unsupported/incorrect statements.

**Clinical expert (Flags hallucination, consistent with the LLM-judge):**

> The original note states "O2sat 100% NRB." The model response fails to accurately interpret the abbreviation and clinical terminology in the note, leading to a misunderstanding. "NRB" refers to a non-rebreather mask (reservoir mask), not "100% on room air" as the model describes. This constitutes a serious misreading of the patient's clinical status.

**Output 2 (Qwen2.5-72B-Instruct, CoT):**

---

The model's reasoning includes:

*"Vital Signs: Temperature 95.2°F (hypothermia), BP 132/43 mmHg (hypotension), HR 74, RR 15, O2 saturation 100% on room air."*

**LLM-judge (Flags hallucination):**

States that the patient had "O2 saturation 100% on room air," whereas the note specifies "O2sat 100% NRB" (non-rebreather), so the room-air claim is unsupported. Also labels BP 132/43 outright as hypotension; this is debatable and not in the input. Hence, the response includes unsupported/incorrect statements.

**Clinical expert (Flags hallucination, consistent with the LLM-judge):**

The note states "O2sat 100% NRB," indicating that the patient is on an NRB mask. The model does not correctly understand the abbreviation and clinical terminology, and incorrectly interprets this as "O2 saturation 100% on room air." In addition, the model's assessment of blood pressure and its normal range is inaccurate: a BP of 132/43 does not fall in the hypotensive range. This reflects a lack of understanding of numerical values and clinical knowledge, leading to serious hallucinations in reasoning.

This case illustrates that, even when a CoT strategy instructs the LLM to analyze the patient's note step by step and generate a detailed reasoning process, the resulting explanation may contain *hallucinated or distorted clinical facts*. In this example, oxygen support was misrepresented (incorrectly inferring room air instead of an NRB mask) and mislabeling blood pressure as hypotension. These errors stem from the weaknesses highlighted by our lexical analysis: misunderstanding of clinical abbreviations ("NRB") and numerical values (vital sign ranges). Such misinterpretations may alter a clinician's perception of patient severity and could lead to incorrect downstream decisions in safety-critical settings.

### A.9 Prompt design and variants

To ensure our findings were not artifacts of specific prompt formulations, we tested four additional prompt variations for both zero-shot and CoT inference using established prompting methodologies. Zero-shot variants differed in formality and constraint specification, while CoT variants used diverse reasoning framings. Complete prompt variations are shown below.

We selected 4 representative models that span different architectures and capabilities to provide a balanced assessment of prompt robustness while maintaining experimental feasibility. As shown in Table S7, the performance decrease with CoT inference was consistent across all prompt variations, confirming that our results represent a systematic limitation.

---

**Method 1**

**Zero-Shot:** "Return your answer in the following format. DO NOT GIVE ANY EXPLANATION:"

**Chain-of-Thought:** "Solve it in a step-by-step fashion, return your answer in the following format, PROVIDE DETAILED ANALYSIS BEFORE THE RESULT:"

---

**Method 2**

**Zero-Shot:** "Provide only the final answer in the format below. Do not include any reasoning or explanation."

**Chain-of-Thought:** "Reason through this task step by step like a clinical expert, then return your final answer in the given format:"

---

**Method 3**

**Zero-Shot:** "Output your answer strictly using the following format. Explanations are not allowed."

**Chain-of-Thought:** "Let's use step by step inductive reasoning, given the clinical nature of the task. Return your final answer in the following format:"

---

**Method 4**

**Zero-Shot:** "Respond using the exact format specified. Do not add any additional commentary."

**Chain-of-Thought:** "Determine the answer to the task given the clinical context in a step by step fashion. Give your final answer in the format below:"

---

**Method 5**

**Zero-Shot:** "Return the final answer exactly as outlined below without providing any justification or reasoning."

**Chain-of-Thought:** "Let's break the task into multiple steps and reason step by step as a clinician would. Then, provide your final answer in the specified format:"

---

Table S7: Performance Gap Between Zero-Shot and Chain-of-Thought: $\Delta$score (%)

| Model | Prompt 1* | Prompt 2 | Prompt 3 | Prompt 4 | Prompt 5 | Overall |
|---|---|---|---|---|---|---|
| Llama3.3-70B | −3.10 (7.8%) | −2.08 (5.28%) | −2.75 (6.81%) | −2.89 (7.25%) | −2.24 (5.61%) | −2.61 (6.55%) |
| Mistral-Large | −3.40 (8.0%) | −3.03 (7.14%) | −3.35 (7.91%) | −2.76 (6.64%) | −2.99 (7.10%) | −3.11 (7.36%) |
| Qwen2.5-72B | −2.70 (6.5%) | −2.24 (5.42%) | −1.47 (3.54%) | −3.27 (8.08%) | −1.94 (4.73%) | −2.32 (5.65%) |
| Athene-V2 | −2.40 (5.8%) | −1.72 (4.16%) | −1.19 (2.87%) | −3.11 (7.63%) | −1.70 (4.13%) | −2.02 (4.92%) |

* Prompt 1 is the original task instruction and also the one used in the main experiments.

## A.10   LLM-as-a-Judge

To investigate LLM failure modes, we employed an LLM-as-a-judge framework implemented with OpenAI-o3 to annotate three binary metrics: hallucination, omission, and incompleteness. For each instance, the LLM judge provided a binary assessment indicating the presence of a failure mode, along with an explanatory statement citing specific examples or rationales. Validation against expert human annotations using the same criteria showed close alignment with the LLM-as-a-judge results.

---

**Prompt of LLM-as-a-Judge for Error Analysis**

**Your Task**

As an experienced medical professional with expertise in clinical documentation, diagnosis, and treatment planning, you will evaluate the accuracy and quality of multiple language model outputs on clinical text tasks. Assess responses for the specified error types and provide binary (0/1) judgments with a short, evidence-based justification for each model response.

**Input Components**

You will receive:

- **Task Instructions:** The clinical task the models were asked to perform, which describes the input information and the task requirements.

- **Input:** The text given to the models, which is sourced from patient records, clinical notes, or other relevant medical documentation.

- **Answer:** The correct answer for this instance, which serves as an important reference (i.e., what diagnosis or decision should be made based on the input).

- **Model Responses:** The responses generated by different language models, which should include the chain-of-thought (CoT) reasoning and a final answer, each with an anonymous identifier (e.g., Model_abc12345).

**Evaluation Criteria**

Evaluate each model response independently for:

**1. Hallucination (Binary: 0 or 1)**

Definition: Any claim in the model response that is not supported by the provided Input or reflects incorrect/non-existent medical facts/knowledge. This includes: fabricated facts; invented numeric values (labs/vitals/doses); unsupported temporal/causal inferences; citing diagnoses/medications/history/guidelines absent from the Input; medically false statements (e.g., wrong contraindications, impossible dose ranges, pathophysiology that contradicts accepted evidence, misuse of units); or reliance on incorrect external knowledge.

- **1 = Hallucination:** The response introduces any unsupported or medically false content—for example, a diagnosis not in the Input, a contraindication stated incorrectly, a guideline claim not evidenced by the Input, or incorrect medical knowledge.

- **0 = No Hallucination:** All relevant claims are both (a) grounded in the Input (verbatim quote or faithful paraphrase) and (b) medically sound (consistent with established clinical knowledge).

**2. Omission (Binary: 0 or 1)**

*Definition:* The model fails to include task-required, clinically relevant facts from the provided Input that are necessary to produce the Expected Output. This includes missing: key indicators for disease diagnosis, key qualifiers (onset, duration, timing, laterality), critical numeric details (dose/unit/frequency/range), severity/stage/grade, abnormal/critical flags, or fields explicitly mandated by the task/schema.

- **1 = Omission:** One or more important and required information is ignored or incompletely reported so that the Expected Output is not fully met. This includes: missing the essential medical information, and over-generalization that drops required specificity (e.g., reporting "antibiotic given" when Input specifies "ceftriaxone 1 g q24h").

- **0 = No Omission:** All clinically significant and task-relevant information from the Input that is necessary to fulfill the Expected Output is present.

---

---

**Clinical LLM Evaluation Prompt Continued**

**3. Incomplete (Binary: 0 or 1)**

*Definition:* The response is truncated or not fully executed, or its reasoning is insufficient to reach the conclusion (e.g., missing steps, unjustified leaps, internal inconsistencies). This includes: (i) output cut off or ending mid-sentence; (ii) required components/fields or mandated format/schema missing or only partially filled; (iii) reasoning gaps where key steps are skipped or the final answer conflicts with its own reasoning; (iv) use of unresolved placeholders ("TBD/.../—") or vague generalities where specifics are required; (v) providing only a subset when the task requires exhaustive listing/coverage.

- **1 = Incomplete:** The reasoning steps of model response are unfinished or incomplete. Any of the above conditions is present.
- **0 = Complete:** The model's reasoning steps is completed and response is finished successfully.

**Output Format**

Provide your evaluation as a JSON object with score and reason for each model. Specifically, the reason should explicitly show the evidence (e.g., raw text within Input) and point out the model errors. Use the model's anonymous identifier as the key:

```
{
  "Model_abc12345": {
    "hallucination": 0 or 1,
    "reason_of_hallucination": "several sentence explanation with
        examples or rationale.",
    "omission": 0 or 1,
    "reason_of_omission": "several sentence explanation with
        examples or rationale.",
    "incomplete": 0 or 1,
    "reason_of_incomplete": "several sentence explanation with
        examples or rationale."
  },
  "Model_def67890": {
    "hallucination": 0 or 1,
    "reason_of_hallucination": "several sentence explanation with
        examples or rationale.",
    "omission": 0 or 1,
    "reason_of_omission": "several sentence explanation with
        examples or rationale.",
    "incomplete": 0 or 1,
    "reason_of_incomplete": "several sentence explanation with
        examples or rationale."
  },
  ...
}
```

**Evaluation Guidelines**

1. **Be Objective and Clinically Relevant:** Your assessment should be grounded in medical knowledge and clinical practices and avoid subjective bias.
2. **Concise Yet Justified Reason:** Each reason should be clear, concise (several sentences), and medically sound, and explicitly show the evidence and score justification.
3. **Ensure Precision:** Ensure your evaluation is precise, consistent, and relevant to the specific clinical task requested.
4. **Independent Evaluation:** Evaluate each model independently, do not compare models to each other, only to the expected output and clinical standards.

Now, please evaluate the provided model outputs using these criteria and return your evaluation in the specified JSON format with scores and reasons for models.

