# OpenReview forum: "Why Chain of Thought Fails in Clinical Text Understanding"
_ICLR.cc/2026/Conference — Submitted to ICLR 2026_

### Official Review · Reviewer_h6y6 · 2025-10-14

**Soundness:** 3
**Presentation:** 3
**Contribution:** 3
**Rating:** 8
**Confidence:** 4

**Summary:**

This paper presents (to my knowledge) the first large-scale, systematic assessment of Chain-of-Thought (CoT) prompting on real clinical text. The authors evaluate 95 LLMs across 87 clinical tasks in 9 languages under harmonized zero-shot vs. CoT protocols, then dig into why CoT underperforms via analyses of reasoning length, medical-concept grounding, lexical signatures, and an error taxonomy adjudicated by an LLM-as-judge with clinician validation. The headline result is clear and somewhat surprising: 86.3% of models do worse with CoT in this setting, with larger drops for weaker models. The paper then shows that longer chains correlate with bigger drops, and that better concept alignment mitigates the damage, while hallucination dominates the failure modes. Overall, this is a careful, comprehensive audit with actionable guidance for safe use of CoT on EHR-style text.

**Strengths:**

Scale & realism. 95 models × 87 tasks from clinical notes/consults across nine languages—rare breadth for clinical NLP.
Clear, robust main result. 86.3% of models drop under CoT; even top models show small but consistent declines—well beyond anecdote.
Mechanistic clarity. The reasoning-length penalty and concept-grounding benefit together tell a coherent story about why clinical CoT fails.

**Weaknesses:**

1.Scope of interventions omitted. By design, the study doesn’t test RAG, self-consistency, or few-shot, which many practitioners would try first; a brief appendix pilot (even on a subset) could bound how much these mitigate the observed drops.
2.Grounding extractor choice. Concept alignment relies on cTAKES; acknowledging its limits on non-English notes (and perhaps adding a multilingual UMLS/MedCAT check) would strengthen the multilingual claim.
3.From correlation to causation. Length and alignment analyses are strong correlates; a small controlled intervention study (e.g., enforced brevity; prompting to cite extracted CUIs) would make the causal story even tighter.

**Questions:**

Mitigation at scale. Do lightweight guardrails (max CoT length; require citing extracted CUIs; numeric-field templates) materially reduce the CoT penalty without heavy RAG/few-shot? Any quick wins from your logs?

Multilingual grounding. Did concept-alignment trends hold uniformly across the nine languages, or were certain languages/tasks outliers?

Task sensitivity. Which task families (NER vs. diagnosis vs. QA) benefit the most from “short-CoT + grounding” constraints? Any per-task ablations to share?

Judge robustness. You report very high PPV. Did you also estimate NPV or inter-rater agreement between clinicians to bound false negatives on subtle omissions?

---

> ### Author Response · Authors · 2025-11-26
> **Response to Reviewer h6y6 1/3**
>
> We appreciate your effort in review and these insightful comments.
>
> Below are our responses.(W=Weakness, Q=Questions, R=Response to)
>
> ---
> **W1:** Scope of interventions omitted. By design, the study doesn’t test RAG, self-consistency, or few-shot, which many practitioners would try first; a brief appendix pilot (even on a subset) could bound how much these mitigate the observed drops.
>
> **RW1:** We appreciate this helpful suggestion. This study aims to investigate the performance and behavior of CoT in a generic and commonly used setting, which is why the original submission focused on zero-shot CoT instead of involving more engineered strategies. We also agree that it is important to evaluate whether more sophisticated reasoning variants can alleviate the observed performance degradation.
>
> In this revision, we add additional experiments and analyses based on reviewers' comments:
>
> **(1) Few-shot with CoT.**
>
> **Method**: We randomly select 5 additional examples (only question and correct answer, named as **QA**) as in-context demonstrations. We then generate corresponding CoT-augmented exemplars (question, CoT analysis, and correct answer, named as **CoT**) using Qwen3-Next-80B-A3B-Thinking, the second-best model among 95 LLMs and best-performing open-source LLM under CoT prompting. Due to time and computational constraints, we focus on 7 representative models and evaluate them on all 87 clinical text tasks under four conditions: QA, CoT, QA (Few-shot), and CoT (Few-shot).
>
> **Result**: As shown in the following table:
>
> | Domain | Model | QA | CoT | Δ (CoT-QA) | QA (Few-shot) | CoT (Few-shot) | Δ(CoT-QA) under Few shot |
> |-------|-------|-------|-------|-------|-------|-------|-------|
> | General | Qwen2.5-7B-Instruct | 31.32 | 30.25 | -1.07 (-3.42%) | 41.6 | 35.47 | -6.13 (-14.74%) |
> | General | Qwen2.5-72B-Instruct | 41.62 | 38.86 | -2.76 (-6.63%) | 50.99 | 49.9 | -1.09 (-2.14%) |
> | General | Llama-3.1-8B-Instruct | 28.98 | 29.4 | 0.42 (1.45%) | 43.54 | 40.99 | -2.55 (-5.86%) |
> | General | Llama-3.1-70B-Instruct | 39.09 | 35.1 | -3.99 (-10.21%) | 50.52 | 50.23 | -0.29 (-0.57%) |
> | General | Athene-V2-Chat | 41.69 | 39.34 | -2.35 (-5.64%) | 50.68 | 50.38 | -0.30 (-0.59%) |
> | Medical | MedGemma-27B-it | 40.79 | 38.2 | -2.59 (-6.35%) | 51.97 | 50.79 | -1.18 (-2.27%) |
> | Medical | HuatuoGPT-o1-72B | 41.01 | 38.15 | -2.86 (-6.97%) | 50.21 | 46.62 | -3.59 (-7.15%) |
>
> (*For better visualization, we provide a figure at the following anonymized link: https://postimg.cc/ppVSbYk8*)
>
>
> **Findings**:
> - Few-shot significantly and consistently improves performance compared with zero-shot across all models.
>
> - Adding CoT for few-shot examples does not yield further gains and still slightly degrades performance relative to few-shot QA.
>
> - For 4/7 LLMs (Llama-3.1-70B-Instruct, Qwen2.5-72B-Instruct, Athene-V2-Chat, and medgemma-27b-it), the magnitude of CoT-induced degradation is generally smaller under few-shot than under zero-shot, indicating that few-shot examples may partially mitigate the negative impact of CoT while preserving most of the benefits of in-context learning.
>
> **(2) Self-consistency.**
>
> **Method**: To efficiently aggregate predictions of self-consistency, we focus on 27 classification tasks, including text classification, NLI, and similarity. For 5 representative models, we compare QA, CoT, and QA (self-consistency) and CoT (self-consistency) (sampling 5 times and aggregating predictions by majority vote). The sampling parameters adopt the model's official recommendation or default temperature=1 and top_p=1.
>
> **Result**: As shown in the following table:
>
> | Domain | Model | QA | CoT | Δ(CoT-QA) in Zero-shot | QA (5 sampling) | CoT (5 sampling) | Δ(CoT-QA) in 5-sampling |
> |-------|-------|-------|-------|-------|-------|-------|-------|
> | General | Qwen2.5-72B-Instruct | 70.49 | 68.58 | -1.91 (-2.71%) | 72.31 | 71.54 | -0.77 (-1.06%) |
> | General | Llama-3.1-70B-Instruct | 68.12 | 66.61 | -1.51 (-2.22%) | 69.59 | 69.62 | 0.03 (0.04%) |
> | General | Athene-V2-Chat | 70.51 | 69.05 | -1.46 (-2.07%) | 72.45 | 72.06 | -0.39 (-0.54%) |
> | Medical | HuatuoGPT-o1-72B | 72.32 | 69.56 | -2.76 (-3.82%) | 72.07 | 71.84 | -0.23 (-0.32%) |
> | General | Qwen3-Next-80B-A3B-Thinking | 73.11 | 73.04 | -0.07 (-0.10%) | 73.77 | 73.19 | -0.58 (-0.79%) |
>
> (*For better visualization, we provide a figure at the following anonymized link: https://postimg.cc/mtcrwMMH*)
>
> **Findings**:
>
> - As expected from ensembling, self-consistency generally improves performance over single prediction (greedy decoding).
>
> - For most models, CoT (self-consistency) still not outperform QA (self-consistency), but it significantly narrows the performance gap between QA and CoT, making CoT behavior more reliable than naive single-chain CoT.

---

> ### Author Response · Authors · 2025-11-26
> **Response to Reviewer h6y6 2/3**
>
> - However, in practice, self-consistency introduces significant computational cost (multiple CoT generations per query) and, in real-world workflows, still requires additional effort to inspect or select among multiple reasoning chains.
>
> **(3) CoT variants.**
>
> In the original submission, we have evaluated five paired QA/CoT prompt variants to validate the generalizability of CoT-induced performance drop. The results show that the performance degradation induced by CoT is consistent across these variants.
>
> **(4) RAG**
>
> As RAG may introduce additional confounding factors (e.g., database quality, retrieval errors, outdated or inconsistent knowledge) and its own failure modes, and a recent study even shows that RAG may degrade LLM's performance on medical tasks [6]. We will take them as important directions for future work rather than the focus of this paper.
>
> Overall, these newly added experiments support the main findings that CoT is not reliably beneficial in real-world clinical text tasks, while also showing that certain enhancements (few-shot prompting, self-consistency) can partially mitigate CoT-induced degradation at the cost of additional complexity and computation. We will incorporate these findings and their implications into the revised results, discussion, and limitations sections to clarify the scope and generality of our conclusions.
>
> Full results and analyses will be reported in the revised manuscript and appendix.
>
> ---
>
> **W2:** Grounding extractor choice. Concept alignment relies on cTAKES; acknowledging its limits on non-English notes (and perhaps adding a multilingual UMLS/MedCAT check) would strengthen the multilingual claim.
>
> **RW2:** Thank you for pointing this out. We chose cTAKES because it is a widely used clinical NLP toolkit with well-documented behavior, which has been used as an effective and reliable tool for medical concept extraction in relevant studies. Its deterministic extraction and normalization to CUIs provide a transparent and reproducible proxy for whether key medical concepts in the input are preserved in the outputs. Currently, our concept-alignment analysis focuses on the English subset, since robust, comparable extraction-and-normalization tools are hard to determine for all nine languages. We will discuss this limitation explicitly in the paper and discuss multilingual extensions (e.g., incorporating tools such as MedCAT or multilingual UMLS mappers) as an important direction for future work.
>
> ---
>
> **W3:** From correlation to causation. Length and alignment analyses are strong correlates; a small controlled intervention study (e.g., enforced brevity; prompting to cite extracted CUIs) would make the causal story even tighter.
>
> **RW3:** We appreciate this insightful suggestion. We agree that explicitly constraining CoT length (e.g., enforcing short chains) or prompting to cite more concepts may be an interesting follow-up. In this study, we did not impose additional length limitations or response content because the control of prompt-level length or response content can itself change model behavior and introduce uncontrolled confounded factors. More importantly, in real clinical applications of LLMs, it is hard to set the maximum CoT length or highlight what we need to specifically mention. Our design aims to mimic typical real-world usage scenarios. We will take these methods of behavior control as interesting directions for our future work.
>
> ---
>
> **Q1:** Mitigation at scale. Do lightweight guardrails (max CoT length; require citing extracted CUIs; numeric-field templates) materially reduce the CoT penalty without heavy RAG/few-shot? Any quick wins from your logs?
>
> **RQ1:** As discussed in RW3, directly adding such constraints may change the model behavior, which will introduce uncontrolled confounded issues. Therefore, in this revised manuscript, we mainly explored relatively generic mitigation strategies: few-shot prompting and self-consistency. As reported in the revision, these methods can partially reduce the CoT penalty and bring CoT performance closer to standard QA.
>
> ---
>
> **Q2:** Multilingual grounding. Did concept-alignment trends hold uniformly across the nine languages, or were certain languages/tasks outliers?
>
> **RQ2:** Because robust clinical concept extraction and normalization tools are hard to find and determine for all nine languages, our concept-alignment analysis currently focuses on the English subset using cTAKES. This avoids introducing additional confounders from heterogeneous extractors. As a result, we currently do not claim cross-language uniformity for concept-alignment trends. We will clarify this limitation more explicitly in the manuscript and point to developing or incorporating multilingual concept-alignment pipelines as future work.
>
> ---

---

> > ### Author Response · Authors · 2025-11-26
> > **Response to Reviewer h6y6 3/3**
> >
> > ---
> >
> > **Q3:** Task sensitivity. Which task families (NER vs. diagnosis vs. QA) benefit the most from “short-CoT + grounding” constraints? Any per-task ablations to share?
> >
> > **RQ3:** Thanks for your helpful suggestion. In the revision, we add a performance analysis of task types. Specifically, we report the performance of four representative general and medical LLMs across the 8 task types. The table below illustrates the average scores and relative changes (CoT-QA / QA, Δ%):
> >
> > | Domain | Model | Text Classification | Normalization and Coding | Semantic Similarity | Natural Language Inference | Event Extraction | Named Entity Recognition | Question Answering | Summarization |
> > | ------- | ------- | -------| -------| -------| -------| -------| -------| -------| -------|
> > | General | GPT-4o | 69.16 / 68.81 (-0.50%) | 8.59 / 7.19 (-16.31%) | 47.90 / 41.85 (-12.63%) | 87.37 / 85.62 (-2.00%) | 29.05 / 23.26 (-19.93%) | 45.27 / 39.23 (-13.36%) | 16.82 / 14.41 (-14.34%) | 33.55 / 29.25 (-12.81%) |
> > | General | Qwen2.5-72B-Instruct | 68.52 / 67.27 (-1.83%) | 4.30 / **5.90** (37.16%) | 47.60 / 44.56 (-6.38%) | 83.44 / 80.04 (-4.08%) | 27.71 / 21.54 (-22.27%) | 38.00 / 36.39 (-4.24%) | 17.76 / 15.64 (-11.92%) | 33.83 / 26.53 (-21.58%) |
> > | General | Llama-3.3-70B-Instruct | 65.27 / 65.13 (-0.21%) | 5.34 / **6.20** (16.07%) | 41.86 / 36.02 (-13.95%) | 83.81 / 83.03 (-0.93%) | 24.23 / 18.18 (-24.96%) | 38.47 / 34.28 (-10.89%) | 16.71 / 15.75 (-5.74%) | 32.82 / 24.47 (-25.45%) |
> > | Medical | MedGemma-27B-it | 66.88 / 65.56 (-1.98%) | 3.31 / **4.12** (24.49%) | 44.65 / 39.56 (-11.39%) | 82.19 / **85.38** (3.89%) | 24.44 / 18.85 (-22.86%) | 40.98 / **41.78** (1.95%) | 16.70 / 8.89 (-46.76%) | 34.13 / 21.40 (-37.29%) |
> > | --- | Overall |  -1.71 (-2.59%) | **0.55** (+11.47%) | -4.13 (-9.47%) | -2.36 (-2.83%) | -5.39 (-21.58%) | -2.61 (-7.10%) | -2.74 (-16.42%) | -6.46 (-20.13%) |
> >
> > Based on these results, we observe that:
> >
> > - CoT degrades LLM performance across most task types, with particularly large relative drops in event extraction, summarization, and question answering.
> >
> > - For text classification and natural language inference, the average impact is smaller but still negative.
> >
> > - Normalization and coding is the only task type where CoT yields a consistent average improvement, albeit starting from relatively low scores, suggesting that structured mapping problems may occasionally benefit from step-by-step reasoning.
> >
> > ---
> >
> > **Q4:** Judge robustness. You report very high PPV. Did you also estimate NPV or inter-rater agreement between clinicians to bound false negatives on subtle omissions?
> >
> > **RQ4:** Thank you for this helpful suggestion. Previously, we focused on PPV as we wanted to capture more error patterns related to CoT failures. We agree that NPV (by annotating cases the judge deems correct) could provide a more complete picture of false negatives and subtle omissions. Due to the time and annotation burden, our clinicians can not complete it during this rebuttal period. We will complete this annotation and analysis in our revised manuscript.
> >
> > ---

---

### Official Review · Reviewer_oUVC · 2025-10-29

**Soundness:** 3
**Presentation:** 3
**Contribution:** 2
**Rating:** 4
**Confidence:** 3

**Summary:**

This paper introduces a study for chain of thought (CoT) effect on medical text understanding. Finding is that CoT degrades performance in most clinical nlp tasks, at an average of 86% of the models being worse than just zero shot, where they benchmarked on 95 models across 87 real world clinical text datasets.

**Strengths:**

- The trade of study on accuracy vs. interpretability for CoT in medicine is timely
- The error taxonomy is well motivated and furthermore the failure modes are validated by clinicians for reliability

**Weaknesses:**

- CoT experiment idea is interesting but it felt the contribution is not substantial. It only compares CoT with zero shot inference, so effectively it's just ask LLMs to 'think step by step'. The novelty of the paper doesn't seem to fully flesh out and a bit over claimed. While the paper’s empirical scale is impressive, the core insight—“CoT reduces accuracy in clinical text tasks”—is primarily observational, not explanatory. The work stops at describing what happens rather than advancing why or how in a mechanistically rigorous way. They gesture at causes (longer reasoning chains, concept misalignment), but these are correlational rather than causal analyses. Mostly “Scale ≠ contribution” evaluating 95 models is extensive, but without a deeper model of reasoning failure, the work risks being seen as descriptive rather than scientific
- The eval metrics don't extend beyond BRIDGE which can be quite limiting. Metrics like accuracy, F1, and ROUGE are purely syntactic; they fail to capture clinical reasoning quality or safety relevance. There’s no human assessment of clinical harm or factual distortion beyond a small 200-sample subset.
- In addition, the metrics are overly quantitative, neglecting qualitative insight where qualitative examples of CoT failures (hallucinations, omissions) are absent from the main text and no detailed case study or sample reasoning trace is shown to illustrate what these failures look like in practice. Figures are mostly aggregate plots (performance trends, correlations) without clinical interpretability examples—a major gap for a paper claiming safety relevance.

**Questions:**

- The baseline (zero-shot) itself is unclear — many tasks (e.g., NER, de-identification) normally require few-shot or schema grounding to perform sensibly
- The Appendix tables list 95 models but omit details of prompt instantiation per task, API parameters, and evaluation scripts

---

> ### Author Response · Authors · 2025-11-26
> **Response to Reviewer oUVC 1/3**
>
> We appreciate your effort in review and insightful comments.
>
> Below are our responses.(W=Weakness, Q=Questions, R=Response)
>
> ---
> **W1:** CoT experiment idea is interesting but it felt the contribution is not substantial. It only compares CoT with zero shot inference, so effectively it's just ask LLMs to 'think step by step'. The novelty of the paper doesn't seem to fully flesh out and a bit over claimed. While the paper’s empirical scale is impressive, the core insight—“CoT reduces accuracy in clinical text tasks”—is primarily observational, not explanatory. The work stops at describing what happens rather than advancing why or how in a mechanistically rigorous way. They gesture at causes (longer reasoning chains, concept misalignment), but these are correlational rather than causal analyses. Mostly “Scale ≠ contribution” evaluating 95 models is extensive, but without a deeper model of reasoning failure, the work risks being seen as descriptive rather than scientific
>
> **RW1:** We appreciate the reviewer’s thoughtful comment and suggestions. Transparent reasoning and better interpretability are critical for applying LLMs in clinical settings. Despite the widespread success of CoT across different domains such as mathematics and coding, the actual usability and effectiveness of CoT in *clinical text* tasks remain largely underexplored. To bridge this gap, our goal is to provide a systematic, clinically grounded characterization of how CoT performs and behaves in real-world clinical text.
>
> Beyond an observational performance comparison, we also aim to move one step closer to understanding *how* and *when* CoT fails by combining several complementary analyses. In addition to the large-scale evaluation reveals that CoT causes widespread performance degradation across LLMs, we analyze model behavior along multiple dimensions: reasoning length, medical concept alignment, lexical signatures, and error distribution.
>
> Taken together, these empirical findings deepen our understanding of LLMs with CoT on clinical text tasks, alarm concrete risks of applying CoT in clinical settings, and suggest actionable directions. For instance, strengthening medical concept alignment, improving the handling of numbers and clinical abbreviations, and using structured error analysis to inform future mitigation strategies. We believe this provides practically relevant insight for designing and deploying LLMs in clinical applications.
>
> We fully agree that strict causal experiments or mechanistic accounts (e.g., probing internal representations, controlled counterfactual edits) would further enhance our understanding of the underlying mechanism. Building on the empirical foundation established here, we will take the causal and mechanistic analyses as an important direction for future work.
>
> In the revision, we also extend our analysis to include few-shot with CoT, self-consistency, and subgroup analyses by task type and language. These additions show that the observed issues are robust to common CoT variants and provide an overview of where explicit reasoning currently helps or hurts in clinical text, further enriching our understanding of CoT for clinical NLP.
>
> ---
>
> **W2:** The eval metrics don't extend beyond BRIDGE which can be quite limiting. Metrics like accuracy, F1, and ROUGE are purely syntactic; they fail to capture clinical reasoning quality or safety relevance. There’s no human assessment of clinical harm or factual distortion beyond a small 200-sample subset.
>
> **RW2:** Thanks for your reminder. In this work, the first step is to establish whether CoT *improves or degrades task performance* on broad clinical text tasks. For this reason, these standard task metrics (accuracy, F1, ROUGE) are required to ensure the fair, objective, and fast evaluation on 95 LLMs across 87 tasks.
>
> Beyond these syntactic metrics, the efficient LLM-as-a-judge was adopted to conduct large-scale error analysis (e.g., hallucination, omission, incomplete reasoning). Meanwhile, we also enrolled clinical experts to perform human annotation on a subset to characterize failure modes, and also validated the reliability of LLM-Judge.
>
> We acknowledged that the expert-reviewed subset is limited in size, largely due to the high annotation burden of reading long clinical notes for each case and carefully assessing their clinical correctness. But the high consistency between human and LLM-Judge can validate the reliability of LLM-Judge. We will clarify this constraint more explicitly in the paper.

---

> ### Author Response · Authors · 2025-11-26
> **Response to Reviewer oUVC 2/3**
>
> ---
>
> **W3:** In addition, the metrics are overly quantitative, neglecting qualitative insight where qualitative examples of CoT failures (hallucinations, omissions) are absent from the main text and no detailed case study or sample reasoning trace is shown to illustrate what these failures look like in practice. Figures are mostly aggregate plots (performance trends, correlations) without clinical interpretability examples—a major gap for a paper claiming safety relevance.
>
> **RW3:** We appreciate this helpful suggestion. In addition to the error analysis supported by LLM-as-a-judge and clinical experts, we now add a case study to provide a more intuitive understanding of how CoT fails in practice.
>
> Case study:
> (*As the used clinical datasets are regulated, we omit most details and rephrase some sentences to protect privacy while preserving the medically relevant structure and errors.*)
>
> - **Task:** MIMIC-III Outcome – in-hospital mortality prediction.
>
> - **Instruction of CoT:**
>
>     >Given a patient's basic information and admission notes, predict the patient's in-hospital mortality, which means whether the patient will die during the current admission.
>     Solve it in a step-by-step fashion, return your answer in the following format, PROVIDE DETAILED ANALYSIS BEFORE THE RESULT:
>     Analysis:
>     ...
>     Result:
>     In-Hospital Mortality: label
>     The optional list for "label" is ["Yes", "No"].
>
> - **Input (excerpt):**
>
>   The clinical note includes vital signs such as:
>   > `PHYSICAL EXAM: VS: T 95.2 BP 132/43 HR 74 RR 15 O2sat 100% NRB`
>   (where “NRB” denotes a non-rebreather mask.)
>
> - **Output 1 (Llama-3.3-70B-Instruct, CoT):**
>
>   In its step-by-step analysis, the model writes:
>   > “The patient's physical exam revealed a temperature of 95.2, blood pressure of 132/43, and oxygen saturation of 100% on room air.”
>
>   **LLM-judge:** Flags hallucination:
>   > States that the patient had “O2 saturation 100% on room air,” whereas the note specifies “O2sat 100% NRB” (non-rebreather), so the room-air claim is unsupported. Also labels BP 132/43 outright as hypotension; this is debatable and not in the input. Hence, the response includes unsupported/incorrect statements.
>
>   **Clinical expert:** Flags hallucination (same as LLM-Judge):
>   > The original note states “O2sat 100% NRB.” The model response fails to accurately interpret the abbreviation and clinical terminology in the note, leading to a misunderstanding. “NRB” refers to a non-rebreather mask (reservoir mask), not “100% on room air” as the model describes. This constitutes a serious misreading of the patient’s clinical status.
>
> - **Output 2 (Qwen2.5-72B-Instruct, CoT):**
>
>   The model’s reasoning includes:
>   > “Vital Signs: Temperature 95.2°F (hypothermia), BP 132/43 mmHg (hypotension), HR 74, RR 15, O2 saturation 100% on room air.”
>
>   **LLM-judge:** Flags hallucination:
>   > States that the patient had “O2 saturation 100% on room air,” whereas the note specifies “O2sat 100% NRB” (non-rebreather), so the room-air claim is unsupported. Also labels BP 132/43 outright as hypotension; this is debatable and not in the input. Hence, the response includes unsupported/incorrect statements.
>
>   **Clinical expert:** Flags hallucination:
>   > The note states “O2sat 100% NRB,” indicating that the patient is on an NRB mask. The model does not correctly understand the abbreviation and clinical terminology, and incorrectly interprets this as “O2 saturation 100% on room air.” In addition, the model’s assessment of blood pressure and its normal range is inaccurate: a BP of 132/43 does not fall in the hypotensive range. This reflects a lack of understanding of numerical values and clinical knowledge, leading to serious hallucinations in reasoning.
>
> - **Discussion.**
>     > This case illustrates that, even when CoT strategy instructs the LLM to analyze the patient’s note step by step and generate a reasoning process, which may contain **hallucinated or distorted clinical facts**, e.g., misrepresenting oxygen support and mislabeling blood pressure. These errors stem from the weaknesses highlighted by our lexical analysis: misunderstanding around **abbreviations** (“NRB”) and **numerical values** (vital sign ranges). Such misinterpretations may fundamentally alter the clinician’s perception of patient severity and result in critical outcomes.
>
> Additionally, we also agree that a separate causal experiment should be helpful, but it also requires strict and controlled counterfactual interventions. We will take it as an interesting and important direction that we plan to pursue in future work.
>
> We will include this case study in the main text and the Appendix.
>
> ---

---

> ### Author Response · Authors · 2025-11-26
> **Response to Reviewer oUVC 3/3**
>
> ---
>
> **Q1:** The baseline (zero-shot) itself is unclear — many tasks (e.g., NER, de-identification) normally require few-shot or schema grounding to perform sensibly
>
> **RQ1:** We appreciate the reviewer for this helpful suggestion. We now add the experiment and analysis of the few-shot setting. Specifically, we randomly select 5 examples (question and correct answer, denoted **QA**) as in-context demonstrations. We then generate corresponding CoT-augmented exemplars (question, CoT analysis, and correct answer, named as **CoT**) using Qwen3-Next-80B-A3B-Thinking, the second-best model among 95 LLMs and best-performing open-source LLM under CoT prompting. Due to time and computational constraints, we focus on 7 representative models and evaluate them on all 87 clinical text tasks under four conditions: QA, CoT, QA (Few-shot), and CoT (Few-shot).
>
> **Result**: As shown in the following table:
>
> | Domain | Model | QA | CoT | Δ (CoT-QA) | QA (Few-shot) | CoT (Few-shot) | Δ(CoT-QA) under Few shot |
> |-------|-------|-------|-------|-------|-------|-------|-------|
> | General | Qwen2.5-7B-Instruct | 31.32 | 30.25 | -1.07 (-3.42%) | 41.6 | 35.47 | -6.13 (-14.74%) |
> | General | Qwen2.5-72B-Instruct | 41.62 | 38.86 | -2.76 (-6.63%) | 50.99 | 49.9 | -1.09 (-2.14%) |
> | General | Llama-3.1-8B-Instruct | 28.98 | 29.4 | 0.42 (1.45%) | 43.54 | 40.99 | -2.55 (-5.86%) |
> | General | Llama-3.1-70B-Instruct | 39.09 | 35.1 | -3.99 (-10.21%) | 50.52 | 50.23 | -0.29 (-0.57%) |
> | General | Athene-V2-Chat | 41.69 | 39.34 | -2.35 (-5.64%) | 50.68 | 50.38 | -0.30 (-0.59%) |
> | Medical | MedGemma-27B-it | 40.79 | 38.2 | -2.59 (-6.35%) | 51.97 | 50.79 | -1.18 (-2.27%) |
> | Medical | HuatuoGPT-o1-72B | 41.01 | 38.15 | -2.86 (-6.97%) | 50.21 | 46.62 | -3.59 (-7.15%) |
>
> (*For better visualization, we provide a figure at the following anonymized link: https://postimg.cc/ppVSbYk8*)
>
> **Findings**:
> - Few-shot significantly and consistently improves performance compared with zero-shot across all models.
>
> - Adding CoT for few-shot examples does not yield further gains and still slightly degrades performance relative to few-shot QA.
>
> - For 4/7 LLMs (Llama-3.1-70B-Instruct, Qwen2.5-72B-Instruct, Athene-V2-Chat, and medgemma-27b-it), the magnitude of CoT-induced degradation is generally smaller under few-shot than under zero-shot, indicating that few-shot examples may partially mitigate the negative impact of CoT while preserving most of the benefits of in-context learning.
>
> ---
>
> **Q2:** The Appendix tables list 95 models but omit details of prompt instantiation per task, API parameters, and evaluation scripts
>
> **RQ2:** Thank you for the reminder. The evaluated tasks are from the BRIDGE benchmark. We followed the official splits, task instructions, and provided the evaluation scripts. The detailed task instructions (base prompts) can be found in the original source.
>
> On top of these task instructions, we apply the QA/CoT wrapper as described in **Appendix A.4 (Prompt design and variants)**. Especially, we define 5 pairs of QA vs. CoT prompts and obtain the final prompt for each run by appending the corresponding QA or CoT template to the original task instruction.
>
> Implementation details, including decoding and API parameters, are summarized in **Appendix A.1 (Implementation details)**. In brief, except for newly added experiments of self-consistency, we use greedy decoding to obtain deterministic predictions (temperature = 0, sampling disabled). For self-consistency, we enable sampling with parameters following the model’s official recommendations when available; otherwise, we use temperature = 1 and top_p = 1 and aggregate 5 samples by majority vote.
>
> We will provide more details about the experiment in our revised manuscript to more explicitly point to these sections.
>
> ---

---

### Official Review · Reviewer_jLqc · 2025-10-31

**Soundness:** 3
**Presentation:** 3
**Contribution:** 2
**Rating:** 4
**Confidence:** 4

**Summary:**

The paper presents the first large-scale systematic investigation into the effectiveness of Chain-of-Thought (CoT) prompting for clinical text understanding, evaluating 95 large language models across 87 real-world, multilingual clinical tasks derived from electronic health records. Contrary to findings in general domains, the study robustly demonstrates that CoT prompting systematically undermines accuracy for the majority of models, particularly lower-capability ones. The authors perform detailed mechanistic analyses showing that performance degradation is linked to increased reasoning length, poor clinical concept grounding, and brittleness in handling numerical data and abbreviations. Finally, an error taxonomy validated by expert clinicians identifies hallucination and omission as the primary failure modes of CoT traces, offering foundational guidance for developing safer and more reliable reasoning protocols in clinical AI.

**Strengths:**

- This work provides an unprecedentedly large-scale and systematic empirical assessment of Chain-of-Thought prompting across diverse clinical text tasks and models.
- The paper demonstrates strong originality by moving beyond traditional medical QA benchmarks to evaluate CoT reliability on complex, noisy, real-world EHR narratives.
- Mechanistic analyses clearly link CoT performance degradation to specific factors like reasoning trace length and weak clinical concept grounding.
- Establishing an error taxonomy (hallucination, omission) supported by LLM-as-a-Judge validation offers a crucial new safety evaluation framework for clinical LLMs.

**Weaknesses:**

- Focusing mainly on zero-shot CoT, the findings do not directly address whether more sophisticated reasoning techniques like few-shot CoT or self-consistency would mitigate the observed performance decline.
- Although acknowledged, the correlational relationship between reasoning trace length and performance decline remains ambiguous regarding causality versus input difficulty.
- The lexical signature analysis identifies potential weaknesses in numerical and abbreviation handling but does not include direct experimental confirmation that these are causal failure mechanisms.
- The authors deliberately exclude practical intervention strategies like RAG, yet these are essential for safety-critical grounding and are likely necessary to overcome the identified hallucination issues.
- Relying on cTAKES for concept alignment metrics, rather than an LLM-based grounding metric, might underestimate or inaccurately characterize complex semantic grounding for higher-capability models.
- The methodology for selecting the three error-audited tasks, out of 87 tasks total, needs clearer justification regarding their representativeness of the overall failure modes or maximal CoT degradation.
- Discussion of the few models (13 out of 95) that showed marginal improvements under CoT is insufficient to provide guidance on conditions that might benefit from explicit reasoning.

**Questions:**

- To clarify the length-performance relationship, could the authors provide results if they experimentally constrained the maximum token length of the Chain-of-Thought reasoning trace?
- Were the observed performance drops uniform across the eight clinical task types, or did certain tasks (e.g., NER, QA, classification) demonstrate disproportionately high or low susceptibility to CoT failure?
- Considering the observed lexical signatures related to numerical brittleness, what results would an experiment yield if the prompt were modified to strictly enforce verbatim quotation of numerical and abbreviated inputs in the reasoning trace?

---

> ### Author Response · Authors · 2025-11-26
> **Response to Reviewer jLqc 1/4**
>
> We appreciate your effort in review and these helpful comments.
>
> Below are our responses.(W=Weakness, Q=Questions, R=Response to)
>
> ---
> **W1:** Focusing mainly on zero-shot CoT, the findings do not directly address whether more sophisticated reasoning techniques like few-shot CoT or self-consistency would mitigate the observed performance decline.
>
> **RW1:** Thanks for this valuable suggestion. As discussed in Section limitations, our primary goal is to investigate the effectiveness of CoT in a general and common setting; therefore, we did not include more specific engineering interventions such as RAG.
> In this revision, to further improve the generality of the main findings, we add additional experiments and analyses based on reviewers' comments:
>
> **(1) Few-shot with CoT.**
>
> **Method**: We randomly select 5 additional examples (only question and correct answer, named as **QA**) as in-context demonstrations. We then generate corresponding CoT-augmented exemplars (question, CoT analysis, and correct answer, named as **CoT**) using Qwen3-Next-80B-A3B-Thinking, the second-best model among 95 LLMs and best-performing open-source LLM under CoT prompting. Due to time and computational constraints, we focus on 7 representative models and evaluate them on all 87 clinical text tasks under four conditions: QA, CoT, QA (Few-shot), and CoT (Few-shot).
>
> **Result**: As shown in the following table:
>
> | Domain | Model | QA | CoT | Δ (CoT-QA) | QA (Few-shot) | CoT (Few-shot) | Δ(CoT-QA) under Few shot |
> |-------|-------|-------|-------|-------|-------|-------|-------|
> | General | Qwen2.5-7B-Instruct | 31.32 | 30.25 | -1.07 (-3.42%) | 41.6 | 35.47 | -6.13 (-14.74%) |
> | General | Qwen2.5-72B-Instruct | 41.62 | 38.86 | -2.76 (-6.63%) | 50.99 | 49.9 | -1.09 (-2.14%) |
> | General | Llama-3.1-8B-Instruct | 28.98 | 29.4 | 0.42 (1.45%) | 43.54 | 40.99 | -2.55 (-5.86%) |
> | General | Llama-3.1-70B-Instruct | 39.09 | 35.1 | -3.99 (-10.21%) | 50.52 | 50.23 | -0.29 (-0.57%) |
> | General | Athene-V2-Chat | 41.69 | 39.34 | -2.35 (-5.64%) | 50.68 | 50.38 | -0.30 (-0.59%) |
> | Medical | MedGemma-27B-it | 40.79 | 38.2 | -2.59 (-6.35%) | 51.97 | 50.79 | -1.18 (-2.27%) |
> | Medical | HuatuoGPT-o1-72B | 41.01 | 38.15 | -2.86 (-6.97%) | 50.21 | 46.62 | -3.59 (-7.15%) |
>
> (*For better visualization, we provide a figure at the following anonymized link: https://postimg.cc/ppVSbYk8*)
>
>
> **Findings**:
> - Few-shot significantly and consistently improves performance compared with zero-shot across all models.
>
> - Adding CoT for few-shot examples does not yield further gains and still slightly degrades performance relative to few-shot QA.
>
> - For 4/7 LLMs (Llama-3.1-70B-Instruct, Qwen2.5-72B-Instruct, Athene-V2-Chat, and medgemma-27b-it), the magnitude of CoT-induced degradation is generally smaller under few-shot than under zero-shot, indicating that few-shot examples may partially mitigate the negative impact of CoT while preserving most of the benefits of in-context learning.
>
> **(2) Self-consistency.**
>
> **Method**: To efficiently aggregate predictions of self-consistency, we focus on 27 classification tasks, including text classification, NLI, and similarity. For 5 representative models, we compare QA, CoT, and QA (self-consistency) and CoT (self-consistency) (sampling 5 times and aggregating predictions by majority vote). The sampling parameters adopt the model's official recommendation or default temperature=1 and top_p=1.
>
> **Result**: As shown in the following table:
>
> | Domain | Model | QA | CoT | Δ(CoT-QA) in Zero-shot | QA (5 sampling) | CoT (5 sampling) | Δ(CoT-QA) in 5-sampling |
> |-------|-------|-------|-------|-------|-------|-------|-------|
> | General | Qwen2.5-72B-Instruct | 70.49 | 68.58 | -1.91 (-2.71%) | 72.31 | 71.54 | -0.77 (-1.06%) |
> | General | Llama-3.1-70B-Instruct | 68.12 | 66.61 | -1.51 (-2.22%) | 69.59 | 69.62 | 0.03 (0.04%) |
> | General | Athene-V2-Chat | 70.51 | 69.05 | -1.46 (-2.07%) | 72.45 | 72.06 | -0.39 (-0.54%) |
> | Medical | HuatuoGPT-o1-72B | 72.32 | 69.56 | -2.76 (-3.82%) | 72.07 | 71.84 | -0.23 (-0.32%) |
> | General | Qwen3-Next-80B-A3B-Thinking | 73.11 | 73.04 | -0.07 (-0.10%) | 73.77 | 73.19 | -0.58 (-0.79%) |
>
> (*For better visualization, we provide a figure at the following anonymized link: https://postimg.cc/mtcrwMMH*)
>
> **Findings**:
>
> - As expected from ensembling, self-consistency generally improves performance over single prediction (greedy decoding).
>
> - For most models, CoT (self-consistency) still not outperform QA (self-consistency), but it significantly narrows the performance gap between QA and CoT, making CoT behavior more reliable than naive single-chain CoT.
>
> - However, in practice, self-consistency introduces significant computational cost (multiple CoT generations per query) and, in real-world workflows, still requires additional effort to inspect or select among multiple reasoning chains.
>
> ---

---

> ### Author Response · Authors · 2025-11-26
> **Response to Reviewer jLqc 2/4**
>
> ---
>
> **W2:** Although acknowledged, the correlational relationship between reasoning trace length and performance decline remains ambiguous regarding causality versus input difficulty.
>
> **RW2:** We appreciate this clarification. In Figure 4, we group test instances into bins according to the observed CoT length and then compare QA vs. CoT performance within each bin. For a given bin, QA and CoT are evaluated on exactly the same set of samples, so input difficulty is held fixed while we examine how the performance gap Δ = (CoT − QA) / QA changes as length increases. We observe that this relative performance drop becomes larger in bins with longer CoT traces, indicating that longer reasoning may be associated with additional degradation.
>
> The reasoning trace length and the performance decline an interesting observations through our large-scale experiments. But we also agree that it does not represent causality, as indicated by the reviewer that the task difficulty is a strong potential cofounder.
>
> We will make this analysis clearer in the revision.
>
> ---
>
> **W3:** The lexical signature analysis identifies potential weaknesses in numerical and abbreviation handling but does not include direct experimental confirmation that these are causal failure mechanisms.
>
> **RW3:** We appreciate this insightful comment. The lexical signature analysis identified potential patterns that are disproportionately associated with success or failures (e.g., numerical expressions and medical abbreviations). But we also agree that it's hard to offer direct experimental confirmation of causal failure mechanisms.
>
> Integrating error detection supported by LLM-Judge and clinical experts, we now add a case study to provide a more intuitive understanding of how CoT fails in practice.
>
> Case study:
> (*As the used clinical datasets are regulated, we omit most details and rephrase some sentences to protect privacy while preserving the medically relevant structure and errors.*)
>
> - **Task:** MIMIC-III Outcome – in-hospital mortality prediction.
>
> - **Instruction of CoT:**
>
>     >Given a patient's basic information and admission notes, predict the patient's in-hospital mortality, which means whether the patient will die during the current admission.
>     Solve it in a step-by-step fashion, return your answer in the following format, PROVIDE DETAILED ANALYSIS BEFORE THE RESULT:
>     Analysis:
>     ...
>     Result:
>     In-Hospital Mortality: label
>     The optional list for "label" is ["Yes", "No"].
>
> - **Input (excerpt):**
>
>   The clinical note includes vital signs such as:
>   > `PHYSICAL EXAM: VS: T 95.2 BP 132/43 HR 74 RR 15 O2sat 100% NRB`
>   (where “NRB” denotes a non-rebreather mask.)
>
> - **Output 1 (Llama-3.3-70B-Instruct, CoT):**
>
>   In its step-by-step analysis, the model writes:
>   > “The patient's physical exam revealed a temperature of 95.2, blood pressure of 132/43, and oxygen saturation of 100% on room air.”
>
>   **LLM-judge:** Flags hallucination:
>   > States that the patient had “O2 saturation 100% on room air,” whereas the note specifies “O2sat 100% NRB” (non-rebreather), so the room-air claim is unsupported. Also labels BP 132/43 outright as hypotension; this is debatable and not in the input. Hence, the response includes unsupported/incorrect statements.
>
>   **Clinical expert:** Flags hallucination (same as LLM-Judge):
>   > The original note states “O2sat 100% NRB.” The model response fails to accurately interpret the abbreviation and clinical terminology in the note, leading to a misunderstanding. “NRB” refers to a non-rebreather mask (reservoir mask), not “100% on room air” as the model describes. This constitutes a serious misreading of the patient’s clinical status.
>
> - **Output 2 (Qwen2.5-72B-Instruct, CoT):**
>
>   The model’s reasoning includes:
>   > “Vital Signs: Temperature 95.2°F (hypothermia), BP 132/43 mmHg (hypotension), HR 74, RR 15, O2 saturation 100% on room air.”
>
>   **LLM-judge:** Flags hallucination:
>   > States that the patient had “O2 saturation 100% on room air,” whereas the note specifies “O2sat 100% NRB” (non-rebreather), so the room-air claim is unsupported. Also labels BP 132/43 outright as hypotension; this is debatable and not in the input. Hence, the response includes unsupported/incorrect statements.
>
>   **Clinical expert:** Flags hallucination:
>   > The note states “O2sat 100% NRB,” indicating that the patient is on an NRB mask. The model does not correctly understand the abbreviation and clinical terminology, and incorrectly interprets this as “O2 saturation 100% on room air.” In addition, the model’s assessment of blood pressure and its normal range is inaccurate: a BP of 132/43 does not fall in the hypotensive range. This reflects a lack of understanding of numerical values and clinical knowledge, leading to serious hallucinations in reasoning.

---

> ### Author Response · Authors · 2025-11-26
> **Response to Reviewer jLqc 3/4**
>
> - **Discussion.**
>     > This case illustrates that, even when CoT strategy instructs the LLM to analyze the patient’s note step by step and generate a reasoning process, which may contain **hallucinated or distorted clinical facts**, e.g., misrepresenting oxygen support and mislabeling blood pressure. These errors stem from the weaknesses highlighted by our lexical analysis: misunderstanding around **abbreviations** (“NRB”) and **numerical values** (vital sign ranges). Such misinterpretations may fundamentally alter the clinician’s perception of patient severity and result in critical outcomes.
>
> Additionally, we also agree that a separate causal experiment should be helpful, but it also requires strict and controlled counterfactual interventions. We will take it as an interesting and important direction that we plan to pursue in future work.
>
> We will include this case study in the main text and the Appendix.
>
> ---
>
> **W4:** The authors deliberately exclude practical intervention strategies like RAG, yet these are essential for safety-critical grounding and are likely necessary to overcome the identified hallucination issues.
>
> **RW4:** We appreciate this insightful comment. We agree that RAG strategies may be helpful for clinical tasks. However, our work aims to characterize the performance and behavior of CoT under generic and commonly used settings, before adding further engineering modules. Additionally, as RAG may introduce additional confounding factors (e.g., database quality, retrieval errors, outdated or inconsistent knowledge) and its own failure modes, and a recent study even shows that RAG may degrade LLM's performance on medical tasks [6]. We will clarify this scope in the revised manuscript and explicitly point to RAG as promising directions.
>
> ---
>
> **W5:** Relying on cTAKES for concept alignment metrics, rather than an LLM-based grounding metric, might underestimate or inaccurately characterize complex semantic grounding for higher-capability models.
>
> **RW5:** We chose cTAKES because it is a widely used clinical NLP toolkit with well-documented behavior, which has also been used as an effective and reliable tool for medical concept extraction in relevant studies. Its deterministic extraction and normalization to CUIs provide a transparent and reproducible proxy for whether key medical concepts in the input are preserved in the model’s outputs. In contrast, LLM-based evaluators may introduce potential hallucination, bias, and domain mismatch, are more computationally demanding, and, as prior work shows, still struggle with reliable clinical coding. We will clarify that our cTAKES-based metric should be viewed as a conservative, reproducible proxy rather than a perfect measure of semantic grounding, and that combining other tools with carefully designed LLM-based evaluators is an interesting direction for future work.
>
> ---
>
> **W6:** The methodology for selecting the three error-audited tasks, out of 87 tasks total, needs clearer justification regarding their representativeness of the overall failure modes or maximal CoT degradation.
>
> **RW6:** Thank you for raising this point. To qualitatively examine typical CoT failure modes with LLM-as-a-judge and expert review, we selected three tasks: mortality prediction on MIMIC-III-Outcome (van Aken et al., 2021) and disease diagnosis on MIMIC-IV-DiReCT (Wang et al., 2024) and MIMIC-IV-CDM (Hager et al., 2024). These tasks (i) are based on real-world EHRs with rich clinical notes, (ii) have clear, clinically meaningful classification labels, and (iii) require integrating long, information-dense inputs—precisely the setting where CoT is expected to help but where hallucination or omission can be most harmful. We will spell out these selection criteria more explicitly in the revision.
>
> ---
>
> **W7:** Discussion of the few models (13 out of 95) that showed marginal improvements under CoT is insufficient to provide guidance on conditions that might benefit from explicit reasoning.
>
> **RW7:** We agree that it is also important to understand when CoT can be beneficial. In our analysis, the 13 models with positive average deltas are heterogeneous (spanning both general and medical models and different sizes), and their gains are generally small and concentrated in specific task–language combinations (e.g., normalization/coding or certain non-English settings). Moreover, these improvements are not robust: they often shrink or disappear in other tasks. We will enrich the discussion to emphasize that, while explicit reasoning may be helpful under some conditions.
>
> ---

---

> ### Author Response · Authors · 2025-11-26
> **Response to Reviewer jLqc 4/4**
>
> ---
>
> **Q1:** To clarify the length-performance relationship, could the authors provide results if they experimentally constrained the maximum token length of the Chain-of-Thought reasoning trace?
>
> **RQ1:** We agree that explicitly constraining CoT length (e.g., enforcing short chains) would be an interesting follow-up. In this study, we did not impose additional length caps, because prompt-level length control can itself change model behavior, and aggressive truncation risks cutting off the final answer. More importantly, in real clinical applications of LLMs, it is uncommon to restrict the maximum CoT length. Our design aims to mimic typical real-world usage scenarios. We may explore constrained-length CoT experiments in future work.
>
> ---
>
> **Q2:** Were the observed performance drops uniform across the eight clinical task types, or did certain tasks (e.g., NER, QA, classification) demonstrate disproportionately high or low susceptibility to CoT failure?
>
> **RQ2:** Thanks for your helpful suggestion. In the revision, we add a performance analysis of task types. Specifically, we report the performance of four representative general and medical LLMs across the 8 task types. The table below illustrates the average scores and relative changes (CoT-QA / QA, Δ%):
>
> | Domain | Model | Text Classification | Normalization and Coding | Semantic Similarity | Natural Language Inference | Event Extraction | Named Entity Recognition | Question Answering | Summarization |
> | ------- | ------- | -------| -------| -------| -------| -------| -------| -------| -------|
> | General | GPT-4o | 69.16 / 68.81 (-0.50%) | 8.59 / 7.19 (-16.31%) | 47.90 / 41.85 (-12.63%) | 87.37 / 85.62 (-2.00%) | 29.05 / 23.26 (-19.93%) | 45.27 / 39.23 (-13.36%) | 16.82 / 14.41 (-14.34%) | 33.55 / 29.25 (-12.81%) |
> | General | Qwen2.5-72B-Instruct | 68.52 / 67.27 (-1.83%) | 4.30 / **5.90** (37.16%) | 47.60 / 44.56 (-6.38%) | 83.44 / 80.04 (-4.08%) | 27.71 / 21.54 (-22.27%) | 38.00 / 36.39 (-4.24%) | 17.76 / 15.64 (-11.92%) | 33.83 / 26.53 (-21.58%) |
> | General | Llama-3.3-70B-Instruct | 65.27 / 65.13 (-0.21%) | 5.34 / **6.20** (16.07%) | 41.86 / 36.02 (-13.95%) | 83.81 / 83.03 (-0.93%) | 24.23 / 18.18 (-24.96%) | 38.47 / 34.28 (-10.89%) | 16.71 / 15.75 (-5.74%) | 32.82 / 24.47 (-25.45%) |
> | Medical | MedGemma-27B-it | 66.88 / 65.56 (-1.98%) | 3.31 / **4.12** (24.49%) | 44.65 / 39.56 (-11.39%) | 82.19 / **85.38** (3.89%) | 24.44 / 18.85 (-22.86%) | 40.98 / **41.78** (1.95%) | 16.70 / 8.89 (-46.76%) | 34.13 / 21.40 (-37.29%) |
> | Overall | LLMs |  -0.76 (-1.13%) | **0.47** (+8.66%) | -5.00 (-10.99%) | -0.68 (-0.81%) | -5.90 (-22.38%) | -2.76 (-6.79%) | -3.32 (-19.56%) | -8.17 (-24.33%) |
>
> (*The bold indicates CoT improves model performance.*)
>
> (*For better visualization, we provide a figure at the following anonymized link: https://postimg.cc/21QT11HV*)
>
> Based on these results, we observe that:
>
> - CoT degrades LLM performance across most task types, with particularly large relative drops in event extraction, summarization, and question answering.
>
> - For text classification and natural language inference, the average impact is smaller but still negative.
>
> - Normalization and coding is the only task type where CoT yields a consistent average improvement, albeit starting from relatively low scores, suggesting that structured mapping problems may occasionally benefit from step-by-step reasoning.
>
> ---
>
> **Q3:** Considering the observed lexical signatures related to numerical brittleness, what results would an experiment yield if the prompt were modified to strictly enforce verbatim quotation of numerical and abbreviated inputs in the reasoning trace?
>
> **RQ3:** We appreciate this thoughtful suggestion. We agree that requiring models to consider numerical values and abbreviations verbatim from the input could help reduce some surface-level hallucinations (e.g., inventing new numbers or mis-typing abbreviations), and prior work has also noted that LLMs may hard to exactly capture the numerical information.
>
> However, as discussed in RW3, many of the failures we observe involve *misinterpretation* or misuse of numeric and abbreviated information (e.g., incorrect thresholds, temporal qualifiers), rather than only incorrect copying. A prompt that enforces verbatim quotation would therefore likely mitigate only part of the problem. Designing and validating such prompts consistently across 87 tasks and 95 models would also introduce additional confounders (e.g., distribution shift due to stricter instructions), which we cannot systematically control within the scope of this work. We will enrich the relevant discussion based on the added case study.
>
> ---

---

> > ### Comment · Reviewer_jLqc · 2025-11-27
> >
> > Thanks for your comprehensive rebuttal, I have raised my scores accordingly.

---

### Official Review · Reviewer_SVPi · 2025-10-31

**Soundness:** 2
**Presentation:** 2
**Contribution:** 1
**Rating:** 2
**Confidence:** 4

**Summary:**

This paper presents the first large-scale, systematic study investigating the efficacy of Chain-of-Thought (CoT) prompting for large language models (LLMs) on clinical text understanding tasks. The authors conduct an exhaustive evaluation across 95 LLMs and 87 real-world clinical tasks, spanning 9 languages and 8 task types. The work argues that CoT can undermine reliability on noisy, fragmented EHR-like inputs and calls for clinically grounded reasoning strategies.

**Strengths:**

- The evaluation scale (95×87, multilingual, multi-task) far exceeds prior clinical CoT reports and sets a solid empirical reference point for the community.

**Weaknesses:**

- The central observation "longer CoT can accumulate errors and degrade performance in noisy settings"  aligns with prior intuition in the hallucination literature. The contribution is primarily scale and domain-specific validation, rather than a new conceptual insight into why CoT fails.

- The paper evaluates only vanilla CoT, without incorporating widely-used practical enhancements such as self-consistency, few-shot prompting, RAG, or structured CoT. In the absence of these comparisons, the work risks overstating the generality of its conclusions.

- The paper aggregates 87 tasks across 8 types but never presents a breakdown of CoT's effect by task type. This is a critical omission. It is highly plausible that tasks like Named Entity Recognition (NER) or Event Extraction do not require multi-step reasoning in the first place. Forcing CoT on such tasks may simply be introducing noise or an inappropriate cognitive pathway, making the resulting performance drop unsurprising and arguably uninformative.

- Although the study claims multilingual evaluation across 9 languages, there is no language-specific analysis.

**Questions:**

- Can you provide the performance breakdown across the 8 task types?
- Do the results consistently hold across all 9 languages?
- Did you try any standard CoT variants (self-consistency, few-shot, retrieval-augmented prompts)?
- Does controlling CoT length or enforcing structured formats change the outcome?

---

> ### Author Response · Authors · 2025-11-26
> **Response to Reviewer SVPi 1/4**
>
> We appreciate your effort in review and these insightful comments.
>
> Below are our point to point responses with multiple responses.(W=Weakness, Q=Questions, R=Response to)
>
> ---
> **W1:** The central observation "longer CoT can accumulate errors and degrade performance in noisy settings" aligns with prior intuition in the hallucination literature. The contribution is primarily scale and domain-specific validation, rather than a new conceptual insight into why CoT fails.
>
> **RW1:** We thank the reviewer for raising this point. While prior work has discussed the hallucination risk associated with Chains-of-Thought (CoT), CoT has also been proven to improve performance in many tasks across different domains, including mathematics and coding [1]. Meanwhile, existing literatures also reports contradictory conclusions regarding LLM with CoT in medical questions [2,3,4,5]. But they primarily focus on medical exams, and the effectiveness of LLM with CoT in clinical text tasks remains largely underexplored: there is not a systematical and large-scale study to investigate whether CoT helps or hurts, and by how much, in real-world clinical text task. This study aims to provide such a comprehensive and clinically grounded characterization of how and why CoT fails in clinical text context.
>
> First, we compare the performance of 95 LLMs on 87 real-world clinical text tasks under both zero-shot and CoT settings, spanning 8 task types and 9 languages, establishing a broad picture of how CoT performs in clinical text contexts. We also examine LLMs' performance across multiple CoT prompt variants, and in the revision, we additionally include self-consistency and few-shot CoT, demonstrating the robustness of our findings (see below).
>
> Beyond performance analysis, we characterize CoT behavior and explore the patterns underlying its successes and failures. Across tasks, we find that longer reasoning traces are frequently associated with performance degradation; that stronger alignment between CoT outputs and key medical concepts in the input tends to mitigate this degradation; that large-scale lexical analyses reveal systematic errors around medical abbreviations and numerical information (e.g., dosages, lab values); and that, based on LLM-as-a-judge along with expert validations, dominant failure modes were further examined, including hallucination, omission of critical information, and incomplete reasoning chains.
>
> Taken together, this study substantially extends our empirical understanding of when and how CoT fails in clinical text and offers several insightful perspective for analyzing CoT behavior in clinical applications. Furthermore, it also offer several meaningful direction，such as strengthening medical concept alignment, improvingthe handling of numbers and clinical abbreviations, and using structured error analysis to inform future mitigation strategies, which We believe will promote the relevant studies about applying LLMs in clinical context.
>
> We will revise the introduction and contribution statements to make this positioning clearer.
>
> [1] Wei, Jason, et al. "Chain-of-thought prompting elicits reasoning in large language models." Advances in neural information processing systems 35 (2022): 24824-24837.
>
> [2] Savage, Thomas, et al. "Diagnostic reasoning prompts reveal the potential for large language model interpretability in medicine." NPJ Digital Medicine 7.1 (2024): 20.
>
> [3] Lucas, Mary M., et al. "Reasoning with large language models for medical question answering." Journal of the American Medical Informatics Association 31.9 (2024): 1964-1975.
>
> [4] Wu, Jiageng, et al. "Large language models leverage external knowledge to extend clinical insight beyond language boundaries." Journal of the American Medical Informatics Association 31.9 (2024): 2054-2064.
>
> [5] Liévin, Valentin, et al. "Can large language models reason about medical questions?." Patterns 5.3 (2024).
>
> ---

---

> ### Author Response · Authors · 2025-11-26
> **Response to Reviewer SVPi 2/4**
>
> **W2:** The paper evaluates only vanilla CoT, without incorporating widely-used practical enhancements such as self-consistency, few-shot prompting, RAG, or structured CoT. In the absence of these comparisons, the work risks overstating the generality of its conclusions.
>
> **RW2:** Thanks for this valuable suggestion. As discussed in Section limitations, our primary goal is to investigate the effectiveness of CoT in a general and common setting; therefore, we did not include more specific engineering interventions such as RAG.
> In this revision, to further improve the generality of the main findings, we add additional experiments and analyses based on reviewers' comments:
>
> **(1) Few-shot with CoT.**
>
> **Method**: We randomly select 5 additional examples (only question and correct answer, named as **QA**) as in-context demonstrations. We then generate corresponding CoT-augmented exemplars (question, CoT analysis, and correct answer, named as **CoT**) using Qwen3-Next-80B-A3B-Thinking, the second-best model among 95 LLMs and best-performing open-source LLM under CoT prompting. Due to time and computational constraints, we focus on 7 representative models and evaluate them on all 87 clinical text tasks under four conditions: QA, CoT, QA (Few-shot), and CoT (Few-shot).
>
> **Result**: As shown in the following table:
>
> | Domain | Model | QA | CoT | Δ (CoT-QA) | QA (Few-shot) | CoT (Few-shot) | Δ(CoT-QA) under Few shot |
> |-------|-------|-------|-------|-------|-------|-------|-------|
> | General | Qwen2.5-7B-Instruct | 31.32 | 30.25 | -1.07 (-3.42%) | 41.6 | 35.47 | -6.13 (-14.74%) |
> | General | Qwen2.5-72B-Instruct | 41.62 | 38.86 | -2.76 (-6.63%) | 50.99 | 49.9 | -1.09 (-2.14%) |
> | General | Llama-3.1-8B-Instruct | 28.98 | 29.4 | 0.42 (1.45%) | 43.54 | 40.99 | -2.55 (-5.86%) |
> | General | Llama-3.1-70B-Instruct | 39.09 | 35.1 | -3.99 (-10.21%) | 50.52 | 50.23 | -0.29 (-0.57%) |
> | General | Athene-V2-Chat | 41.69 | 39.34 | -2.35 (-5.64%) | 50.68 | 50.38 | -0.30 (-0.59%) |
> | Medical | MedGemma-27B-it | 40.79 | 38.2 | -2.59 (-6.35%) | 51.97 | 50.79 | -1.18 (-2.27%) |
> | Medical | HuatuoGPT-o1-72B | 41.01 | 38.15 | -2.86 (-6.97%) | 50.21 | 46.62 | -3.59 (-7.15%) |
>
> (*For better visualization, we provide a figure at the following anonymized link: https://postimg.cc/ppVSbYk8*)
>
>
> **Findings**:
> - Few-shot significantly and consistently improves performance compared with zero-shot across all models.
>
> - Adding CoT for few-shot examples does not yield further gains and still slightly degrades performance relative to few-shot QA.
>
> - For 4/7 LLMs (Llama-3.1-70B-Instruct, Qwen2.5-72B-Instruct, Athene-V2-Chat, and medgemma-27b-it), the magnitude of CoT-induced degradation is generally smaller under few-shot than under zero-shot, indicating that few-shot examples may partially mitigate the negative impact of CoT while preserving most of the benefits of in-context learning.
>
> **(2) Self-consistency.**
>
> **Method**: To efficiently aggregate predictions of self-consistency, we focus on 27 classification tasks, including text classification, NLI, and similarity. For 5 representative models, we compare QA, CoT, and QA (self-consistency) and CoT (self-consistency) (sampling 5 times and aggregating predictions by majority vote). The sampling parameters adopt the model's official recommendation or default temperature=1 and top_p=1.
>
> **Result**: As shown in the following table:
>
> | Domain | Model | QA | CoT | Δ(CoT-QA) in Zero-shot | QA (5 sampling) | CoT (5 sampling) | Δ(CoT-QA) in 5-sampling |
> |-------|-------|-------|-------|-------|-------|-------|-------|
> | General | Qwen2.5-72B-Instruct | 70.49 | 68.58 | -1.91 (-2.71%) | 72.31 | 71.54 | -0.77 (-1.06%) |
> | General | Llama-3.1-70B-Instruct | 68.12 | 66.61 | -1.51 (-2.22%) | 69.59 | 69.62 | 0.03 (0.04%) |
> | General | Athene-V2-Chat | 70.51 | 69.05 | -1.46 (-2.07%) | 72.45 | 72.06 | -0.39 (-0.54%) |
> | Medical | HuatuoGPT-o1-72B | 72.32 | 69.56 | -2.76 (-3.82%) | 72.07 | 71.84 | -0.23 (-0.32%) |
> | General | Qwen3-Next-80B-A3B-Thinking | 73.11 | 73.04 | -0.07 (-0.10%) | 73.77 | 73.19 | -0.58 (-0.79%) |
>
> (*For better visualization, we provide a figure at the following anonymized link: https://postimg.cc/mtcrwMMH*)
>
> **Findings**:
>
> - As expected from ensembling, self-consistency generally improves performance over single prediction (greedy decoding).
>
> - For most models, CoT (self-consistency) still not outperform QA (self-consistency), but it significantly narrows the performance gap between QA and CoT, making CoT behavior more reliable than naive single-chain CoT.
>
> - However, in practice, self-consistency introduces significant computational cost (multiple CoT generations per query) and, in real-world workflows, still requires additional effort to inspect or select among multiple reasoning chains.

---

> ### Author Response · Authors · 2025-11-26
> **Response to Reviewer SVPi 3/4**
>
> **(3) CoT variants.**
>
> In the original submission, we have evaluated five paired QA/CoT prompt variants to validate the generalizability of CoT-induced performance drop. The results show that the performance degradation induced by CoT is consistent across these variants.
>
> **(4) RAG and Structured CoT**
>
> As RAG may introduce additional confounding factors (e.g., database quality, retrieval errors, outdated or inconsistent knowledge) and its own failure modes, and a recent study even shows that RAG may degrade LLM's performance on medical tasks [6]. Meanwhile, structured CoT methods (e.g., ToT[7] or GoT[8]) are promising but rely on substantial engineering, which is not yet standard practice in routine clinical applications. We will take them as important directions for future work rather than the focus of this paper.
>
> Overall, these newly added experiments support the main findings that CoT is not reliably beneficial in real-world clinical text tasks, while also showing that certain enhancements (few-shot prompting, self-consistency) can partially mitigate CoT-induced degradation at the cost of additional complexity and computation. We will incorporate these findings and their implications into the revised results, discussion, and limitations sections to clarify the scope and generality of our conclusions.
>
> Full results and analyses will be reported in the revised manuscript and appendix.
>
> [6] Kim, Hyunjae, et al. "Rethinking Retrieval-Augmented Generation for Medicine: A Large-Scale, Systematic Expert Evaluation and Practical Insights." arXiv preprint arXiv:2511.06738 (2025).
>
> [7] Yao, Shunyu, et al. "Tree of thoughts: Deliberate problem solving with large language models." Advances in neural information processing systems 36 (2023): 11809-11822.
>
> [8] Besta, Maciej, et al. "Graph of thoughts: Solving elaborate problems with large language models." Proceedings of the AAAI conference on artificial intelligence. Vol. 38. No. 16. 2024.
>
> ---
> **W3:** The paper aggregates 87 tasks across 8 types but never presents a breakdown of CoT's effect by task type. This is a critical omission. It is highly plausible that tasks like Named Entity Recognition (NER) or Event Extraction do not require multi-step reasoning in the first place. Forcing CoT on such tasks may simply be introducing noise or an inappropriate cognitive pathway, making the resulting performance drop unsurprising and arguably uninformative.
>
> **RW3:** We appreciate this helpful suggestion. The overall score across 87 tasks could summarize how CoT performs across broad real-world clinical text tasks. We agree that a subgroup analysis of task type can help us understand where CoT is most (un)helpful, and we have added such an analysis in the revision.
>
> Specifically, we report the performance of four representative general and medical LLMs across the 8 task types. The table below illustrates the average scores and relative changes (CoT-QA / QA, Δ%):
>
> | Domain | Model | Text Classification | Normalization and Coding | Semantic Similarity | Natural Language Inference | Event Extraction | Named Entity Recognition | Question Answering | Summarization |
> | ------- | ------- | -------| -------| -------| -------| -------| -------| -------| -------|
> | General | GPT-4o | 69.16 / 68.81 (-0.50%) | 8.59 / 7.19 (-16.31%) | 47.90 / 41.85 (-12.63%) | 87.37 / 85.62 (-2.00%) | 29.05 / 23.26 (-19.93%) | 45.27 / 39.23 (-13.36%) | 16.82 / 14.41 (-14.34%) | 33.55 / 29.25 (-12.81%) |
> | General | Qwen2.5-72B-Instruct | 68.52 / 67.27 (-1.83%) | 4.30 / **5.90** (37.16%) | 47.60 / 44.56 (-6.38%) | 83.44 / 80.04 (-4.08%) | 27.71 / 21.54 (-22.27%) | 38.00 / 36.39 (-4.24%) | 17.76 / 15.64 (-11.92%) | 33.83 / 26.53 (-21.58%) |
> | General | Llama-3.3-70B-Instruct | 65.27 / 65.13 (-0.21%) | 5.34 / **6.20** (16.07%) | 41.86 / 36.02 (-13.95%) | 83.81 / 83.03 (-0.93%) | 24.23 / 18.18 (-24.96%) | 38.47 / 34.28 (-10.89%) | 16.71 / 15.75 (-5.74%) | 32.82 / 24.47 (-25.45%) |
> | Medical | MedGemma-27B-it | 66.88 / 65.56 (-1.98%) | 3.31 / **4.12** (24.49%) | 44.65 / 39.56 (-11.39%) | 82.19 / **85.38** (3.89%) | 24.44 / 18.85 (-22.86%) | 40.98 / **41.78** (1.95%) | 16.70 / 8.89 (-46.76%) | 34.13 / 21.40 (-37.29%) |
> | Overall | LLMs |  -0.76 (-1.13%) | **0.47** (+8.66%) | -5.00 (-10.99%) | -0.68 (-0.81%) | -5.90 (-22.38%) | -2.76 (-6.79%) | -3.32 (-19.56%) | -8.17 (-24.33%) |
>
> (*The bold indicates CoT improves model performance.*)
>
> (*For better visualization, we provide a figure at the following anonymized link: https://postimg.cc/21QT11HV*)
>
> Based on these results, we observe that:
>
> - CoT degrades LLM performance across most task types, with particularly large relative drops in event extraction, summarization, and question answering.
>
> - For text classification and natural language inference, the average impact is smaller but still negative.

---

> ### Author Response · Authors · 2025-11-26
> **Response to Reviewer SVPi 4/4**
>
> - Normalization and coding is the only task type where CoT yields a consistent average improvement, albeit starting from relatively low scores, suggesting that structured mapping problems may occasionally benefit from step-by-step reasoning.
>
> Regarding the necessity of CoT for NER and event extraction tasks, clinical information extraction is often context-dependent and nuanced, involving phenomena such as temporality (current vs. historical conditions), negation, uncertainty, and cross-sentence linkage. In such cases, CoT explanations can in principle help users understand *why* the model extracted certain spans and how it interpreted the surrounding context. And, our results also demonstrate that CoT did not impair model performance more on these tasks than on other task types.
>
> ---
>
> **W4:** Although the study claims multilingual evaluation across 9 languages, there is no language-specific analysis.
>
> **RW4:** Thanks for your helpful suggestion. In this revision, we now provide a language-specific subgroup analysis of CoT’s impact. The table below shows the average performance of 4 representative LLMs across the 9 languages, comparing QA vs. CoT:
>
> | Domain | Model | English | Portuguese | Spanish | Chinese | Japanese | Russian | French | German | Norwegian |
> | ------- | ------- | -------| -------| -------| -------| -------| -------| -------| -------| -------|
> | General | gpt-4o | 47.19 / 43.10 (-8.68%) | 57.69 / **58.65** (1.67%) | 25.40 / 22.82 (-10.15%) | 43.86 / 41.19 (-6.08%) | 37.40 / 30.82 (-17.61%) | 64.71 / 63.98 (-1.13%) | 48.82 / 38.71 (-20.72%) | 52.16 / 46.81 (-10.26%) | 54.15 / 53.57 (-1.07%) |
> | General | Qwen2.5-72B-Instruct | 45.54 / 42.76 (-6.09%) | 61.26 / 53.66 (-12.41%) | 23.75 / 20.29 (-14.59%) | 41.71 / 39.44 (-5.43%) | 34.48 / 31.59 (-8.38%) | 59.55 / 59.18 (-0.61%) | 50.02 / 40.01 (-20.01%) | 42.09 / **44.08** (4.73%) | 38.19 / **39.11** (2.40%) |
> | General | Llama-3.3-70B-Instruct | 44.60 / 39.30 (-11.90%) | 58.66 / 53.59 (-8.64%) | 21.83 / **21.88** (0.22%) | 37.56 / 37.14 (-1.12%) | 30.96 / 25.78 (-16.73%) | 58.79 / **59.93** (1.95%) | 45.56 / 29.59 (-35.06%) | 47.19 / **47.49** (0.64%) | 36.38 / **40.03** (10.04%) |
> | Medical | medgemma-27b-it | 44.62 / 39.18 (-12.19%) | 59.37 / 53.29 (-10.25%) | 20.84 / **21.57** (3.48%) | 39.83 / 37.30 (-6.35%) | 34.22 / 32.94 (-3.76%) | 61.40 / **64.05** (4.32%) | 48.26 / 41.97 (-13.03%) | 48.84 / **54.22** (11.02%) | 41.14 / **43.28** (5.21%) |
> | Overall | LLMs |  -4.40 (-9.68%) | -4.45 (-7.51%) | -1.32 (-5.74%) | -1.97 (-4.84%) | -3.99 (-11.63%) | **0.68** (+1.11%) | -10.60 (-22.00%) | **0.58** (+1.22%) | **1.53** (+3.61%) |
>
> (*For better visualization, we provide a figure at the following anonymized link: https://postimg.cc/K4tKSt4v*)
>
> Based on these results, we observe that:
>
> - CoT-induced degradation are common across different languages. With relative drops of about 3–18% in English, Portuguese, Spanish, Chinese, and Japanese, and an even larger average drop in French (often exceeding 20%). Russian and German are nearly neutral (–1.2% and –0.3%), and Norwegian is the only language with a modest average improvement (+3.1%).
>
> - The magnitude and even the direction of CoT's impact vary by both language and model. But as different languages have different tasks and then the performance variance should not directly compared across languages.
>
> Finally, the language-specific analysis confirms that our main conclusion that CoT is not reliably beneficial on real-world clinical text, which holds across multiple languages, while also highlighting model- and language-dependent variability.
>
> We will incorporate this analysis and discussion into the revised manuscript, thereby strengthening and clarifying our multilingual claims.
>
> ---
>
> **Q1**: Can you provide the performance breakdown across the 8 task types?
>
> **RQ1**: Discussed in RW3.
>
> ---
>
> **Q2**: Do the results consistently hold across all 9 languages?
>
> **RQ2**: Discussed in RW4.
>
> ---
>
> **Q3**: Did you try any standard CoT variants (self-consistency, few-shot, retrieval-augmented prompts)?
>
> **RQ3**: Discussed in RW2.
>
> ---
>
> **Q4**: Does controlling CoT length or enforcing structured formats change the outcome?
>
> **RQ4**: Explicitly constraining CoT length (e.g., enforcing short chains) would be an interesting follow-up experiment. In this study, we did not impose additional length caps, because prompt-level length control can itself change model behavior, and aggressive truncation risks cutting off the final answer. More importantly, in real clinical applications of LLMs, it is hard to restrict the maximum CoT length to a reasonable range. Our design aims to mimic typical real-world usage scenarios. Regarding to structured format, it may require more specific prompt optimization and may introduce other confounded factors. Currently, we do not involve them in our study and will try to explore such methods in our subsequent work.
>
> ---

---

### Author Response · Authors · 2025-12-02
**Summary of rebuttal discussion**

Dear Reviewers, ACs, and PCs,

We sincerely appreciate all your efforts in reviewing our work and these helpful suggestions! These suggestions have greatly helped us further improve our manuscript.

Reviewers’ concerns mainly focused on: novelty, CoT's impact under more settings, and subgroup analyses. These are all addressable, and we have added new experiments and analyses accordingly. Encouragingly, the reviewer also recognized our updates and raised the score.

CoT provides transparent reasoning that is critical for safely applying LLMs in clinical settings, yet its effectiveness in clinical text remains largely unexplored. To address this gap, we conduct a large-scale, systematic evaluation of 95 LLMs on 87 real-world clinical text tasks and analyze their behavior to uncover potential patterns of CoT failure/success. Our work substantially extends our understanding of when, how, and why CoT fails in clinical text, and provides practical directions for improving LLMs deployment in clinical practice.

During this rebuttal, we have made the following updates:

1. **CoT + Few-shot.**
   Added experiments on 7 models across all 87 tasks, comparing standard QA and CoT under zero-shot and few-shot settings. The results show that few-shot consistently improves performance, and that CoT with few-shot still tends to slightly degrade performance (See Appendix A.4, Page 16-17).

2. **CoT + Self-consistency with sampling.**
   Added self-consistency experiments on 27 classification tasks for 5 large LLMs, showing that such a strategy further improves model performance and largely narrows (but does not reverse) the performance gap between QA and CoT, at the cost of higher computational overhead (See Appendix A.5, Page 17-18).

3. **Subgroup analyses across 8 task types and 9 languages.**
   Added detailed analyses across task types and languages, showing that CoT-induced degradation is widespread but heterogeneous, with clear model– and language–dependent variation and normalization/coding as a notable exception (See Appendix A.6 and A.7, Page 18-21).

4. **Case study of CoT failures.**
   Included a case study illustrating hallucinations and misinterpretation of vital signs (abbreviations and numerical ranges), grounding our lexical analysis in concrete clinical errors (See Appendix A.6 and A.8, Page 22-23).

5. **Refine the relevant description and enrich the discussion.**
   We will clarify the scope and novelty of our work in our final manuscript, which highlights limitations more explicitly and strengthens the discussion of implications for safe and reliable use of CoT in clinical applications.

These updates have been incorporated into the updated PDF (mainly in the appendix P16-23 and the section titles highlighted in blue), and we hope to further streamline them into the main text in the final version.




Sincerely,

The Authors

---

### Meta-Review · Area_Chair_Ajxo · 2026-01-05

**Summary:**

This paper performs a large-scale empirical evaluation of Chain-of-Thought (CoT) reasoning on clinical tasks. The observation is that CoT often degrades model performance in such settings, especially for smaller models. The main reservation expressed by reviewers about this work is that the contribution is modest: The original manuscript showed that (vanilla, zero-shot) CoT can degrade performance on clinical tasks. Multiple reviewers pointed out that strategies like self-consistency might mitigate this; the authors did provide some new results in revision to address this.

I'd also highlight that some of the framing seems a bit off to me; specifically around CoT offering "interpretability" (e.g., in the abstract the claim is that "CoT enhances interpretability"). In fact, a fair amount of work suggests one should not take CoT at face value as it is not necessarily faithful. See, e.g., "CoT is not Explainability" which surveys this work.

**Reviewer Concerns:**

Multiple reviewers highlighted that augmenting the CoT strategy (e.g., with self-consistency, amongst other strategies) might change the results. The reviewers did include some additional results, but these are not comprehensive.

The authors also provided results broken down by task type, as requested.

Critiques about the fundamental scope of the contribution here are not addressed.

**Reviewer Scores:**

SVPi asked for augmented CoT methods, including self-consistency; the authors did provide this, although this is not comprehensive. SVPi also requested results broken down by task type, which the authors have provided. They may have adjusted their score upwards marginally.

jLqc and oUVC similarly asked for self-consistency results, so might have bumped scores slightly; jLqc requested several additional clarifications of analyses which the authors appear to have provided, while oUVC raised more conceptual issues with the contribution.

h6y6 was already quite positive.

---

### Decision · Program_Chairs · 2026-01-26

Reject